# Optimal Algorithms for Stochastic Contextual Preference Bandits

**Aadirupa Saha**[*]

## Abstract

We consider the problem of preference bandits in the contextual setting. At each round, the learner is presented with a context set of $K$ items, chosen randomly from a potentially infinite set of arms $\mathcal{D} \subseteq \mathbb{R}^d$. However, unlike classical contextual bandits, our framework only allows the learner to receive feedback in terms of item preferences: At each round, the learner is allowed to play a subset of size $q$ (any $q \in \{2, \ldots, K\}$) upon which only a (noisy) winner of the subset is revealed. Yet, same as the classical setup, the goal is still to compete against the best context arm at each round. The problem is relevant in various online decision-making scenarios, including recommender systems, information retrieval, tournament ranking–typically any application where it's easier to elicit the items' relative strength instead of their absolute scores. To the best of our knowledge, this work is the first to consider preference-based stochastic contextual bandits for potentially infinite decision spaces. We start with presenting two algorithms for the special case of pairwise preferences ($q = 2$): The first algorithm is simple and easy to implement with an $\tilde{O}(d\sqrt{T})$ regret guarantee, while the second algorithm is shown to achieve the optimal $\tilde{O}(\sqrt{dT})$ regret, as follows from our $\Omega(\sqrt{dT})$ matching lower bound analysis. We then proceed to analyze the problem for any general $q$-subsetwise preferences ($q \geq 2$), where surprisingly, our lower bound proves the fundamental performance limit to be $\Omega(\sqrt{dT})$ yet again, independent of the subsetsize $q$. Following this, we propose a matching upper bound algorithm justifying the tightness of our results. This implies having access to subsetwise preferences does not help in faster information aggregation for our feedback model. All the results are corroborated empirically against existing baselines.

## 1 Introduction

Sequential decision-making problems with side information (in the form of features or attributes), have been popular in machine learning as contextual bandits [16, 13, 26]. A contextual bandit learner, at each round, observes a context before taking action based on it. The resulting payoff is typically assumed to depend on the context and the action taken according to an unknown map. The learner aims to play the best possible action for the current context at each time and thus minimize its regret with respect to an oracle that knows the payoff function.

In many learning settings, however, it is more common to be able to only *relatively* compare actions, in a decision step, instead of being able to gauge their absolute utilities. E.g., information retrieval, search engine optimization, recommender systems, crowdsourcing, drug testing, tournament ranking, social surveys etc. [18, 21]. A specific application example is: Consider the problem of recommending items from a catalog to users on a shopping website. Each time, the context is determined by the visiting user's features together with all items' features. When a subset of items is presented, the user clicks on one of them according to a relative preference model; only the items presented matter – this plausibly models comparative cognitive choices being made by humans. The aim is to converge to identify the overall best item in the catalog. Additionally, since our model is designed to leverage

---

[*]Microsoft Research, New York, US; `aasa@microsoft.com`.

the structured preference feedback, we can handle very large decision spaces, unlike state of the art dueling bandits algorithms that only deals with pairwise preferences [47], [23] [43] [41] etc., or even some of the recent works on general subsetwise preference bandits (e.g. [30, 33, 38]). All of them need to maintain a $K^2$ matrix explicitly, and their regret scales as $O(K)$. This becomes impractical for large (or infinite) action spaces of size $K$.

In this work, we consider a natural *structured contextual preference bandit* setting, comprised of items with intrinsic (absolute) scores depending on their features in an unknown way, e.g., linear with unknown weights. In the most general setup, the learner plays (compares) a subset $2 \leq q \leq K$ items and gets to see a (noisy) $\ell$-length rank-ordered feedback (where $1 \leq \ell \leq q$) of the top-ranked items of the selected subset with a probability distribution governed by the items' scores. We are primarily interested in developing adaptive subset-selection algorithms for which guarantees can be given for a suitably defined measure of regret.

To the best of our knowledge, we are the first to give *theoretical guarantees* for the above problem of *regret minimization in contextual preference bandits for potentially infinitely large decision spaces* and design provably optimal algorithms for the same. Some recent works [17, 38] have considered this problem in the subset selection setup but their algorithms do not guarantee any finite time regret bounds, neither validate their performance optimality theoretically. In [15], authors considered a version of adversarial contextual dueling bandit problem, which only takes into account the special case of pairwise feedback ($q = 2$). Though similar in names, their problem framework and consequently the analysis are very different from ours. In their setup, at each time the preference matrix is determined by a randomly chosen known context variable, and the algorithm is assumed to have access to a pool of finite set of policies. The goal of the learner is to compete with the 'optimal policy in the pool' for which they proposed an sparring-EXP4 like algorithm—their regret bound though scales as $O(\sqrt{T})$ in $T$, it depends linearly on the size of arm space ($K$ in their notation), and thus become vacuous for infinitely large decision space. Moreover, their computational complexity also scales linearly with the size of the policy class, which tends to be prohibitively large in practice.

**1.** We propose the problem of structured contextual preference bandits, where at each iteration, the learner is presented with a set of $K$-arms $\mathcal{S}_t$ (each represented by $d$-dimensional feature vectors), and the task of the learner is to select a subset of at most $q$-items ($1 \leq q \leq K$), with the objective being to identify the 'best arm' of $\mathcal{S}_t$ in every round. The novelty of the framework lies in the relative preference based feedback model, which only allows the learner to see a noisy draw of the top-ranked items of the selected subset, unlike the absolute reward feedback used for standard bandits setup.

**2.** We first address the problem for the special case of dueling bandits, where $q = 2$, i.e. the learner only have access to pairwise preferences of the selected item pairs at each round. We propose two algorithms for the basic dueling bandit setting: Our first algorithm, Maximum-Informative-Pair (Alg 1), is based on the idea of selecting the most uncertain pair ('max-variance') from the set of 'promising candidates', and we prove an $O(d\sqrt{T})$ regret for the same (Thm. 3).

**3.** Our second algorithm Stagewise-Adaptive-Duel (Alg. 3), is developed on the idea of tracking, in a phased fashion, the best arm of the context set, which ensures a sharper concentration rate of the pairwise scores. This results in $\tilde{O}(\sqrt{dT})$ [2] regret guarantee (Thm. 5), improving upon the regret bound of our previous algorithm by a $\sqrt{d}$ factor.

**4.** We also show that fundamental regret lower bound of $\Omega(\sqrt{dT})$ for the contextual dueling bandit problem addressed here (Thm. 3). Thus theoretically our second algorithm (Alg. 3) is provably optimal, however our Alg. 1 often works better in practice as we show in the experiments (Sec. 6).

**5.** We then analyze the problem for more general subsetwise preference feedback, where at each round, the learner is allowed to play a subset of $q$-items ($q \geq 2$), upon which the winner feedback of the played subset is revealed [10, 30, 33]. We first prove an $\Omega(\sqrt{dT})$ regret lower bound (Thm. 11) for the problem, which establishes that, interestingly, having access to subsetwise preferences does not really help in faster information aggregation (for the specific preference model considered here). Subsequently, we also discuss an algorithm with near-optimal regret guarantee, whose regret bound is also independent of the subsetsize $q$ up to logarithmic factors (Thm. 12).

---

[2]The notation $\tilde{O}(\cdot)$ hides logarithmic dependencies.

Finally, we corroborate our theoretical findings with empirical evaluations. Detailed *Related works*, *Experiments* and most of the technical *proofs* are moved to the appendix.

**Related Works.** Surprisingly, following the same spirit of extending standard multiarmed bandits (MAB) to continuous decision spaces (as in *linear* or *GP-bandits*), there has been really very little work on the continuous extension of the Dueling Bandit problem [24]. The works in [38, 17] did attempt a similar objective, however, without any satisfactory theoretical performance guarantees. [24] considers the problem of dueling bandits on continuous arm set but under rather restrictive sets of assumptions: Twice continuously differentiable, lipschitz, strongly convex and smooth score/reward function, which are often impractical for modeling any real-world preference feedback. Recently, [28] consider the problem of $k$-way assortment selection, where the problem is to minimize regret against the set of highest revenue. However, their objective is much different from ours. We focus on the regret against the single best item per iteration. Thus our pairwise action set must allow repeated item pulls, unlike their setup, due to which their algorithm does not lead to sublinear regret for our objective. In another recent work, [10] did address the problem of regret minimization in continuous *Dueling Bandits*, however without any finite time regret guarantee of their proposed algorithms, unlike ours. A more elaborated survey is given in Appendix A.

## 2 Preliminaries and Problem Formulation

**Notations.** For any positive integer $n \in \mathbb{N}_+$, we denote by $[n]$ the set $\{1, 2, ..., n\}$. $\mathbf{1}(\varphi)$ is generically used to denote an indicator variable that takes the value 1 if the predicate $\varphi$ is true, and 0 otherwise. The decision space is denoted by $\mathcal{D} \subseteq \mathbb{R}^d$, where $d \in \mathbb{N}_+$. We use $\mathbf{1}_d$ to denote an $d$-dimensional vector of all 1's. For any matrix $M \in \mathbb{R}^{d \times d}$, we denote respectively by $\lambda_{\max}(M)$ and $\lambda_{\min}(M)$ the maximum a minimum eigenvalue of matrix $M$. For any $\mathbf{x} \in \mathbb{R}^d$, $\|\mathbf{x}\|_M := \sqrt{x^\top M \mathbf{x}}$ denotes the weighted $\ell_2$-norm associated with matrix $M$ (assuming $M$ is positive-definite).

### 2.1 Problem Setup

We consider the stochastic $K$-armed contextual dueling bandit problem for $T$ rounds, where at each round $t \in [T]$, the learner is presented with a context set $\mathcal{S}_t = \{\mathbf{x}_1^t, \mathbf{x}_2^t, \ldots, \mathbf{x}_K^t\} \subseteq \mathcal{D} \subset \mathbb{R}^d$ of size $K$ which is drawn IID from some $d$-dimensional decision space $\mathcal{D}$ (according to some unknown distribution on $\mathcal{D}$, say $\mathcal{P}_\mathcal{D}$). The learner is permitted to play a subset $\mathcal{X}_t \subseteq \mathcal{S}_t$ of size $q \geq 2$, given a fixed $q \leq K$ (see the formal setup in Sec. 2.1). Clearly for $q = 2$, the problem reduces to the sequential duel (pair of items) selection, say in this case we denote $\mathcal{X}_t = \{\mathbf{x}_t, \mathbf{y}_t\}$. Upon this, the environment provides a stochastic subsetwise preference feedback as follows:

**Subsetwise-Preference Feedback Model.** At any round $t$, upon selecting $\mathcal{X}_t$, the learner receives a *winner feedback* $o_t$ such that: $Pr\big(o_t = \mathbf{x} \mid \mathcal{X}_t\big) = \frac{e^{g(\mathbf{x})}}{\sum_{\mathbf{y} \in \mathcal{X}_t} e^{g(\mathbf{y})}}$ for any $\mathbf{x} \in \mathcal{X}_t$, where $g : \mathcal{D} \mapsto [0, 1]$ is a utility score function on each point in the decisions space $\mathbf{x} \in \mathcal{D}$. Note our preference model essentially boils down to the well studied Plackett Luce (PL) choice model with the individual PL score of item-$x$ being $e^{g(\mathbf{x})}$ [21, 35, 32, 30]. If $q = 2$, then $o_t = \mathbf{1}(\mathbf{x}_t$ preferred over $\mathbf{y}_t)$ simply indicates the preferred arm of the duel $(\mathbf{x}_t, \mathbf{y}_t)$, such that for any $\mathbf{x}, \mathbf{y} \in \mathcal{D}$, the probability $\mathbf{x}$ is preferred over $\mathbf{y}$, denoted by $Pr(\mathbf{x} \succ \mathbf{y})$, is drawn according to $\sim Ber\big(\sigma(h(\mathbf{x}, \mathbf{y}))\big)$, here $\sigma(\cdot)$ being the sigmoid transformation $\big($i.e. $\sigma(x) = \frac{1}{1+e^{-x}}$ for any $x \in \mathbb{R}\big)$.

**Analysis with linear scores.** In this paper, we assume that $g(\mathbf{x}) = \mathbf{x}^\top \boldsymbol{\theta}^*$, $\forall \mathbf{x} \in \mathcal{D}$, where $\boldsymbol{\theta}^* \in \mathbb{R}^d$ is some unknown fixed vector in $\mathbb{R}^d$ such that $\|\boldsymbol{\theta}^*\| \leq 1$. We will henceforth denote this linear utility based 'subsetwise preference model' as SPM$(\boldsymbol{\theta}^*, d, q)$.

**Objective: Regret Minimization.** Suppose $\mathbf{x}_t^* := \arg\max_{\mathbf{x} \in \mathcal{S}_t} \mathbf{x}^\top \boldsymbol{\theta}^*$ is the best arm (with highest score) of round $t$. Then the goal of the learner is to minimize the $T$-round cumulative regret $R_T = \sum_{t=1}^T r_t$ with respect to the best arm $\mathbf{x}_t^*$ of each round $t$, where we measure the instantaneous regret $r_t$ of playing a set $\mathcal{X}_t$ in terms of the average score of the played duel $\frac{\sum_{\mathbf{x} \in \mathcal{X}_t} \mathbf{x}^\top \boldsymbol{\theta}^*}{|\mathcal{X}_t|}$. Precisely,

$$R_T = \sum_{t=1}^T \left( \mathbf{x}_t^{*\top} \boldsymbol{\theta}^* - \frac{\sum_{\mathbf{x} \in \mathcal{X}_t} \mathbf{x}^\top \boldsymbol{\theta}^*}{|\mathcal{X}_t|} \right). \tag{1}$$

Above notion of learner's regret is motivated from the definition of classical $K$-*armed dueling bandit regret* introduced by [44] which is later adopted by the dueling bandit literature [46, 23, 3, 43, 45, 38, 31]. Here the context set at any round $t$ is assumed to be a fixed set of $K$ arms $\mathcal{S}_t = [K]$, and at each round the instantaneous regret incurred by the learner for playing an arm-pair $(i_t, j_t) \in [K \times K]$ is given by $r_t^{(\text{DB})} = \frac{\mathbf{P}(i_*,i_t)+\mathbf{P}(i_*,j_t)-1}{2}$, $i_* \in [K]$ being the 'best-arm' in the hindsight (e.g. condorcet winner [46] or copeland winner [23, 41]) depending on the underlying preference matrix $\mathbf{P} \in [0,1]^{K \times K}$.

**Remark 1** (Equivalence with Dueling Bandit Regret when $q = 2$). *It is easy to note that assuming the context set $\mathcal{S}_t \subseteq \mathcal{D}$ to be fixed $\forall t \in [T]$ and denoting $\mathbf{x}_t^* = \mathbf{x}^*$, our regret definition (Eqn. (1)) is equivalent to dueling bandit regret (up to constant factors), as in our case the pairwise advantage against the best arm (i.e. $Pr(\mathbf{x}^*, \mathbf{x}_t) - \frac{1}{2}$) can be both upper and lower bounded as:*

*(1)* $Pr(\mathbf{x}^*, \mathbf{x}_t) - \frac{1}{2} = \frac{(e^{\mathbf{x}^{*\top}\boldsymbol{\theta}^*} - e^{\mathbf{x}_t^\top \boldsymbol{\theta}^*})}{2(e^{\mathbf{x}^{*\top}\boldsymbol{\theta}^*} + e^{\mathbf{x}_t^\top \boldsymbol{\theta}^*})} \le \frac{(\mathbf{x}^* - \mathbf{x}_t)^\top \boldsymbol{\theta}^*}{2}$ *and (2)* $Pr(\mathbf{x}^*, \mathbf{x}_t) - \frac{1}{2} \ge \frac{(\mathbf{x}^* - \mathbf{x}_t)^\top \boldsymbol{\theta}^*}{4e}$.

*Combining above claims we get $\frac{R_T}{4e} \le R_T^{(DB)} \le \frac{R_T}{2}$ (analysis detail given in Appendix B.1).*

# 3   Dueling Feedback ($q = 2$): Algorithm and Analysis

We first analyze the problem for pairwise preference feedback ($q = 2$ case). Before proceeding to the actual algorithms, it is crucial to note that, same as generalized linear Bandits (GLB) [16, 26], both our algorithms use standard MLE techniques for maintaining a 'tight estimate' of $\boldsymbol{\theta}^*$ (Line 5 of Alg. 1, Line 8 of Alg. 3). This is since our dueling preference feedback can be seen as a generalized linear reward over item pairs (details in Appendix C). However, the regret definition of dueling bandits (Eqn. (1)) being very different than GLB objective, a direct application of any GLB algorithm will simply lead to $O(T)$ regret in our setup: The objective of any GLB learner is to converge to the arm with highest reward, unlike ours. Thus any GLM routine would always converge to the best-worst arm pair as that would be perceived to be the duel with highest reward (pairwise preference in our case). On the contrary, to have any sublinear regret, we require the learner to eventually play only the best arm in the duel, which does not have the highest pairwise-preference (reward). This is the inherent complexity and primary difference of any dueling bandit problem w.r.t. GLB objective or any MAB framework per se. As a result, a sublinear MAB algorithm never works for dueling/preference bandit objectives. To circumvent the problem, we design fairly non-trivial arm-selection rules for our proposed algorithms, e.g., Alg. 1 first needs to construct a promising-set $\mathcal{C}_t$ and then pick the maximum informative pair amongst them (Line 6-7, Alg1). Alg. 2 needs to optimally maintain the set of 'good-items' $\mathcal{G}^s$ with a careful arm-selection rule which significantly differs from any GLB approach (Line 19, Alg 2). Consequently, we also need to resolve to new proof ideas towards analyzing their regret guarantees which remains one of the primary novelty of this work.

## 3.1   Algorithm-1: Maximum-Informative-Pair

Our first algorithm is a computationally efficient one with a $O(d\sqrt{T})$ regret guarantee (Thm. 3), which is only suboptimal by a factor of $O(\sqrt{d})$ (as reflects from our lower bound, Thm. 10, Sec. 4).

**Main Idea:** At any time $t$, the algorithm simply maintains an UCB estimate on the pairwise scores $\bar{h}(\mathbf{x}, \mathbf{y}) := \hat{\boldsymbol{\theta}}^\top (\mathbf{x} - \mathbf{y}) + \eta \|\mathbf{x} - \mathbf{y}\|_{V_t^{-1}}$ for any pair of arms $(\mathbf{x}, \mathbf{y})$, $\mathbf{x}, \mathbf{y} \in \mathcal{S}_t$, where $V_t = \sum_{\tau=1}^{t-1}(\mathbf{x}_\tau - \mathbf{y}_\tau)(\mathbf{x}_\tau - \mathbf{y}_\tau)^\top$, $\hat{\boldsymbol{\theta}}$ being the maximum likelihood estimator (MLE) of our preference model parameter. It then builds a set of promising arms $\mathcal{C}_t := \{\mathbf{x} \in \mathcal{S}_t \mid \bar{h}(\mathbf{x}, \mathbf{y}) > 0, \forall \mathbf{y} \in \mathbf{y} \in S_t \setminus \{\mathbf{x}\}\}$: Arms that beats the rest in terms of their UCB score $\bar{h}(\mathbf{x}, \mathbf{y})$. Finally it plays the most-uncertain (least sampled) pair $(\mathbf{x}_t, \mathbf{y}_t) := \arg\max_{\mathbf{x}, \mathbf{y} \in \mathcal{C}_t} \|\mathbf{x} - \mathbf{y}\|_{V_t^{-1}}$. Note $(x_t, y_t)$ is the pair with highest pairwise score variance in $\mathcal{C}_t$, hence 'maximum informative'. (Detail in Alg. 1.)

**Analysis.** Regret guarantee of Alg.1 (Thm. 3) is based on the following main lemmas.

**Lemma 1** (Self-Normalized Bound). *Suppose $\{(\mathbf{x}_1, \mathbf{y}_1), (\mathbf{x}_2, \mathbf{y}_2), \ldots, (\mathbf{x}_t, \mathbf{y}_t)\}$ be a sequence of arm-pair played such that all arms $\mathbf{x} \in \{\mathbf{x}_\tau, \mathbf{y}_\tau\}_{\tau=1}^{t}$ belong to the ball of unit radius. Also suppose the initial exploration length $t_0$ be such that $\lambda_{\min}\left(\sum_{\tau=1}^{t_0}(\mathbf{x}_\tau - \mathbf{y}_\tau)(\mathbf{x}_\tau - \mathbf{y}_\tau)^\top\right) \ge 1$. Then $\forall t > t_0$,*

$$\sum_{\tau=t_0+1}^{t} \|(\mathbf{x}_\tau - \mathbf{y}_\tau)\|_{V_{\tau+1}^{-1}} \le \sqrt{2dt \log\left(\frac{4t_0 + t}{d}\right)}, \text{ where recall } V_{\tau+1} := \sum_{j=1}^{\tau}(\mathbf{x}_j - \mathbf{y}_j)(\mathbf{x}_j - \mathbf{y}_j)^\top.$$

---
**Algorithm 1 Maximum-Informative-Pair (MaxInP)**
---
1: **input:** Learning rate $\eta > 0$, exploration length $t_0 > 0$
2: **init:** Select $t_0$ pairs $\{(\mathbf{x}_\tau, \mathbf{y}_\tau)\}_{\tau \in [t_0]}$, each drawn at random from $\mathcal{S}_\tau$, and observe the corresponding preference feedback $\{o_\tau\}_{\tau \in [t_0]}$
3: Set $V_{t_0+1} := \sum_{\tau=1}^{t_0}(\mathbf{x}_\tau - \mathbf{y}_\tau)(\mathbf{x}_\tau - \mathbf{y}_\tau)^\top$
4: **for** $t = t_0 + 1, t_0 + 2, \ldots T$ **do**
5:     Compute the MLE $\hat{\boldsymbol{\theta}}_t$ on $\{(\mathbf{x}_\tau, \mathbf{y}_\tau, o_\tau)\}_{\tau=1}^{t-1}$: $\sum_{\tau=1}^{t-1}\left(o_\tau - \sigma\big((\mathbf{x}_\tau - \mathbf{y}_\tau)^\top \hat{\boldsymbol{\theta}}_t\big)\right)(\mathbf{x}_\tau - \mathbf{y}_\tau) = \mathbf{0}$
6:     $\mathcal{C}_t := \{\mathbf{x}, \mathbf{y} \in \mathcal{S}_t \mid (\mathbf{x} - \mathbf{y})^\top \hat{\boldsymbol{\theta}}_t + \eta\|(\mathbf{x} - \mathbf{y})\|_{V_t^{-1}} > 0\}$
7:     Compute $(\mathbf{x}_t, \mathbf{y}_t) := \arg\max_{\mathbf{x}, \mathbf{y} \in \mathcal{C}_t} \|\mathbf{x} - \mathbf{y}\|_{V_t^{-1}}$
8:     Play the duel $(\mathbf{x}_t, \mathbf{y}_t)$. Receive $o_t = \mathbf{1}(\mathbf{x}_t \text{ beats } \mathbf{y}_t)$
9:     Update $V_{t+1} = V_t + (\mathbf{x}_t - \mathbf{y}_t)(\mathbf{x}_t - \mathbf{y}_t)^\top$
10: **end for**
---

**Lemma 2** (Confidence Ellipsoid). *Suppose the initial exploration length $t_0$ be such that $\lambda_{\min}\left(\sum_{\tau=1}^{t_0}(\mathbf{x}_\tau - \mathbf{y}_\tau)(\mathbf{x}_\tau - \mathbf{y}_\tau)^\top\right) \geq 1$, and $\kappa$ is as defined in Thm. 3. Then for any $\delta > 0$, with probability at least $(1 - \delta)$, for all $t > t_0$, $\|\boldsymbol{\theta}^* - \hat{\boldsymbol{\theta}}_t\|_{V_t} \leq \frac{1}{2\kappa}\sqrt{\frac{d}{2}\log\left(1 + \frac{2t}{d}\right) + \log\frac{1}{\delta}}$, where recall $V_{t+1} := \sum_{\tau=1}^{t}(\mathbf{x}_\tau - \mathbf{y}_\tau)(\mathbf{x}_\tau - \mathbf{y}_\tau)^\top$.*

**Theorem 3** (Regret bound of Maximum-Informative-Pair (Alg. 1)). *Let $\eta = \frac{1}{2\kappa}\sqrt{\frac{d}{2}\log(1 + \frac{2T}{d}) + \log\frac{1}{\delta}}$, where $\kappa := \inf_{\|x-y\|\leq 2, \|\boldsymbol{\theta}^*-\hat{\boldsymbol{\theta}}\|\leq 1}\left[\sigma'\big((\mathbf{x} - \mathbf{y})^\top \hat{\boldsymbol{\theta}}\big)\right]$ is the minimum slope of the estimated sigmoid when $\hat{\boldsymbol{\theta}}$ is sufficiently close to $\boldsymbol{\theta}^*$ $\left(\sigma'(\cdot)\text{ being the first order derivative of the sigmoid function }\sigma(\cdot)\right)$. Then given any $\delta > 0$, with probability at least $(1 - 2\delta)$, the $T$ round cumulative regret of Maximum-Informative-Pair satisfies:*

$$R_T \leq t_0 + \left(\frac{1}{\kappa}\sqrt{\frac{d}{2}\log\left(1 + \frac{2T}{d}\right) + \log\frac{1}{\delta}}\right)\sqrt{2dT\log\left(\frac{4t_0 + T}{d}\right)} = O\left(d\sqrt{T}\log\left(\frac{T}{d\delta}\right)\right),$$

*where we choose $t_0 = 2\left(\frac{C_1\sqrt{d} + C_2\sqrt{\log(1/\delta)}}{\lambda_{\min}(B)}\right)^2 + \frac{4}{\lambda_{\min}(B)}$, $B = \mathbf{E}_{\mathbf{x}, \mathbf{y} \overset{iid}{\sim} \mathcal{P}_\mathcal{D}}[(\mathbf{x} - \mathbf{y})(\mathbf{x} - \mathbf{y})^\top]$ (for some universal problem independent constants $C_1, C_2 > 0$).*

*Proof.* **(sketch)** Our choice of $t_0$ ensures that with probability at least $(1 - \delta)$, $V_{t_0+1}$ is full rank, or more precisely $\lambda_{\min}(V_{t_0+1}) \geq 1$ [42] (see Lem. 13, Appendix D for the formal statement). We next apply the two key concentration lemmas (Lem 1 and 2), upon expressing the regret definition in terms of the above concentration results. Precisely, using Lem. 2 and our *'most informative pair'* based arm selection strategy, we can show at any round $t > t_0$, we can bound $r_t \leq 4\|\mathbf{x}_t - \mathbf{y}_t\|_{V_t^{-1}}$. The results now follows from the choice of $\eta$ and Lem.1. The complete proof is given in Appendix D.1. $\qquad\square$

### 3.2  Algorithm-2: Stagewise-Adaptive-Duel (Sta′D))

Our second algorithm runs with a provable optimal regret bound of $\tilde{O}(\sqrt{dT})$, except with an additional $\sqrt{\log K}$ factor. When $K = O(1)$, the algorithm thus yields an optimal regret guarantee.

**Main Idea.** The algorithm proceeds in stages $s \in \lfloor \log T \rfloor$ with the aim of tracking a set of *'promising arms'* $\mathcal{G}^s$ per stage: At each such stage $s$, we maintain confidence interval on the pairwise scores of each index pair $(i, j)$ $p_t^s(i, j)$. If at any stage $s$, the confidence-score of any arm-pair is not estimated to the *'sufficient accuracy'*, we play that pair and include it in the set of *'informative pairs'* of stage $\phi^s$ to be further explored in following rounds. Otherwise, we sequentially eliminate the *'weakly-performing'* arms which gets defeated by some other arm in terms of its optimistic pairwise score, and proceed to the next stage $s + 1$ to examine the surviving arms with a stricter confidence interval. Now if the pairwise scores of every index pair in the set of *'promising-arms'* $\mathcal{G}^s$ is almost *'accurately estimated up to high confidence'*, we pick the first arm $\mathbf{x}_t$ as the one which has

the maximum estimated score, followed by choosing its strongest challenger $\mathbf{y}_t$ which beats $\mathbf{x}_t$ with highest pairwise preference. The algorithm is given in Alg. 3 (Appendix 3).

**Analysis.** Thm. 5 proves an optimal $\tilde{O}(\sqrt{dT})$ regret bound of Alg. 3 (matching the regret lower bound, Thm. 10). It is worth pointing that the near optimal regret analysis of Stagewise-Adaptive-Duel crucially relies on the stronger concentration guarantees of the pairwise scores (compared to the weaker concentration of Lem. 2 used earlier for Alg. 1). Note, this is made possible by specifically maintaining the independent *'stage-wise informative samples'* $\phi^s$ as also pioneered in few of the earlier works [8, 13] for multi-armed bandits (see our proof analysis of Lem. 4).

**Lemma 4** (Sharper Concentration of Pairwise Scores). *Consider any $\delta > 0$, and suppose we set the parameters of Stagewise-Adaptive-Duel (Alg. 3) as $\eta = \frac{3}{2\kappa}\sqrt{2\log\frac{3TK}{\delta}}$, where $\kappa :=$*
$\inf_{\|\mathbf{x}-\mathbf{y}\|\leq 2, \|\boldsymbol{\theta}^*-\hat{\boldsymbol{\theta}}\|\leq 1}\left[\sigma'\big((\mathbf{x}-\mathbf{y})^\top\hat{\boldsymbol{\theta}}\big)\right]$, *and* $t_0 = 2\left(\frac{C_1\sqrt{d}+C_2\sqrt{\log(2/\delta)}}{\lambda_{\min}(B)}\right)^2 + \frac{4\Lambda}{\lambda_{\min}(B)}$, *where* $\Lambda =$
$\frac{8}{\kappa^4}\big(d^2 + \log\frac{3}{\delta}\big)$ *and* $B = \mathbf{E}_{\mathbf{x},\mathbf{y}\overset{iid}{\sim}\mathcal{P}_{\mathcal{D}}}[(\mathbf{x}-\mathbf{y})(\mathbf{x}-\mathbf{y})^\top]$ *(for some universal problem independent constants $C_1, C_2 > 0$). Then with probability at least $(1-\delta)$, for all stages $s \in \lceil\log T\rceil$ at all rounds $t > t_0$ and for all index pairs $i, j \in \mathcal{G}^s$ of round $t$: $|(\mathbf{x}_i^t - \mathbf{x}_j^t)^\top(\theta^* - \theta_t^s) \leq p_t^s(i,j)|$.*

**Theorem 5** (Regret bound of Stagewise-Adaptive-Duel (Alg. 3)). *Consider we set $t_0$ and $\eta$ as per Lem. 4. Then for any $\delta > 0$, with probability at least $(1-\delta)$, the regret of Alg. 3 can be bounded as:*

$$R_T \leq t_0 + 4\eta\sqrt{2d\log\left(\frac{4t_0 T}{d}\right)}\sqrt{T\log T} + 2\sqrt{T} = O\left(\frac{\sqrt{dT\log T}}{\kappa}\sqrt{\log\left(\frac{TK}{\delta}\right)\log\left(\frac{Td}{\kappa}\log\frac{1}{\delta}\right)}\right)$$

*Proof.* **(sketch)** Suppose we denote by $\phi^c := \{t \in [T] \setminus [t_0] \mid t \notin \cup_{s=1}^{\lfloor\log T\rfloor}\phi^s\}$ the set of all good time intervals where all the index pairs $p_t^s(i,j)$ are estimated within the confidence accuracy $\frac{1}{\sqrt{T}}$. The proof crucially relies on the concentration bound of Lem. 4, from which we first derive:

**Lemma 6.** *For any $t > t_0$, suppose the pair $(\mathbf{x}_t, \mathbf{y}_t)$ is chosen at stage $s_t$, and $i_t^*$ denotes the best action of round $t$, i.e. $\mathbf{x}_{i_t^*}^t = \mathbf{x}_t^* = \arg\max_{\mathbf{x}\in\mathcal{S}_t}\mathbf{x}^\top\boldsymbol{\theta}^*$. Then for any $\delta \in (0,1)$, with probability at least $(1-\delta)$, for all $t > t_0$: $i_t^* \in \mathcal{G}^{s_t}$ and for both $\mathbf{x} \in \{\mathbf{x}_t, \mathbf{y}_t\}$, $g(\mathbf{x}_t^*)-g(\mathbf{x}) \leq \begin{cases} \frac{2}{\sqrt{T}} & \text{if } t \in \phi^c \\ \frac{4}{2^{s_t}} & \text{otherwise} \end{cases}$.*

Owning to Lem. 1 and due to the construction of our *'stagewise-good item pairs'* we can also show:

**Lemma 7.** *At any stage $s \in \lfloor\log T\rfloor$, with probability at least $(1-\delta)$, $\sqrt{|\phi^s|} \leq \eta 2^s\sqrt{2d\log\left(\frac{4t_0 T}{d}\right)}$.*

The final regret bound can be derived combining the results of Lem. 6 and 7. (see Appendix E.5). $\square$

## 4   Lower Bound for Dueling ($q = 2$) Feedback

We now proceed to understand the fundamental performance limit of our contextual preference bandits problem for pairwise preference ($q = 2$) case. Towards this we use a novel idea of reducing linear bandits problem to our setup which finally leads to the desired lower bound of $\Omega(\sqrt{dT})$.

**Reducing Linear-Contextual Bandits to our framework.** Let us instantiate any instance of our $K$-armed contextual dueling bandit problem by its problem parameter $\boldsymbol{\theta}^* \in R^d$ as $\mathcal{I}^{cdb}(\boldsymbol{\theta}^*, K, T)$. On the other hand define any instance of $K$-armed contextual linear bandit problem [13] with problem parameter $\boldsymbol{\theta}^* \in R^d$ as $\mathcal{I}^{clb}(\boldsymbol{\theta}^*, K, T)$: Recall in this setup, at each iteration the learner is provided with a context set $\mathcal{S}_t = \{\mathbf{x}_1^t, \mathbf{x}_2^t, \ldots, \mathbf{x}_K^t\} \subset \mathbb{R}^d$ of size $K$ (as before $\|\mathbf{x}\|_2 \leq 1$, $\forall \mathbf{x} \in \mathcal{S}_t$), upon which the learner choose an arm $\mathbf{x}_t \in \mathcal{S}_t$, and the environment feedbacks a reward $r(\mathbf{x}_t) = \mathbf{x}_t^\top\boldsymbol{\theta}^* + \varepsilon_t$, where $\varepsilon_t$ is a zero mean random noise. Objective is to minimize the regret with respect to the best action, $\mathbf{x}_t^* := \arg\max_{\mathbf{x}\in\mathcal{S}_t}\mathbf{x}^\top\boldsymbol{\theta}^*$, of each round $t$, defined as: $R_T^{clb} := \sum_{t=1}^T\big(\mathbf{x}_t^{*\top}\boldsymbol{\theta}^* - \mathbf{x}_t^\top\boldsymbol{\theta}^*\big)$,

**Main Idea.** For proving a lower bound for $\mathcal{I}^{cdb}(\boldsymbol{\theta}^*, K, T)$, we first show under *Gumbel noise* [9, 34], any instance of contextual linear bandits $\mathcal{I}^{clb}$ can be reduced to an instance of $\mathcal{I}^{cdb}$.

**Lemma 8** (Reducing $\mathcal{I}^{clb}$ with *Gumbel noise* to $\mathcal{I}^{cdb}$). *There exists a reduction from the $\mathcal{I}^{clb}$ problem (under Gumbel noise, i.e. $\varepsilon_t \overset{iid}{\sim} Gumbel(0,1)$) to $\mathcal{I}^{cdb}$ which preserves the expected regret.*

*Proof.* **(sketch)** Suppose we have a blackbox algorithm for the instance of $\mathcal{I}^{cdb}$ problem, say $\mathcal{A}^{cdb}$. To prove the claim, our goal is to show that this can be used to solve the $\mathcal{I}^{clb}$ problem where the underlying stochastic noise, $\epsilon_t$ at round $t$, is generated from a Gumbel$(0, 1)$ distribution [39, 9]. Precisely we can construct an algorithm for $\mathcal{I}^{clb}(\boldsymbol{\theta}^*, K, T)$ (say $\mathcal{A}^{clb}$) using $\mathcal{A}^{cdb}$ as follows:

**Algorithm 2** $\mathcal{A}^{clb}$ for problem $\mathcal{I}^{clb}(\boldsymbol{\theta}^*, K, T)$

1: **for** $t = 1, 2, \ldots \lceil \frac{T}{2} \rceil$ **do**
2:     Receive: $(\mathbf{x}_t, \mathbf{y}_t) \leftarrow$ duel played by $\mathcal{A}^{cdb}$ at time $t$.
3:     Play $\mathbf{x}_t$ at round $(2t - 1)$ of $\mathcal{I}^{clb}$. Receive $r(\mathbf{x}_t)$.
4:     Play $\mathbf{y}_t$ at round $2t$ of $\mathcal{I}^{clb}$. Receive $r(\mathbf{y}_t)$.
5:     Feedback: $o_t = \mathbf{1}(r(\mathbf{x}_t) > r(\mathbf{y}_t))$ to $\mathcal{A}^{cdb}$.
6: **end for**

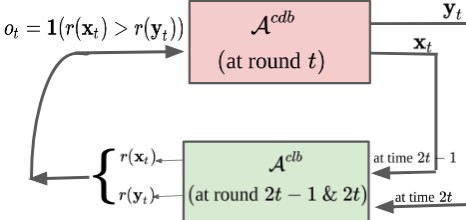

Figure 1: Demonstration of the reduction idea: $\mathcal{I}^{clb}$ to $\mathcal{I}^{cdb}$

**Lemma 9.** *If $\mathcal{A}^{clb}$ rums on a problem instance $\mathcal{I}^{clb}(\boldsymbol{\theta}^*, K, 2T)$ with Gumbel$(0, 1)$ noise, then the internally the algorithm $\mathcal{A}^{cdb}$ runs on a problem instance of $\mathcal{I}^{cdb}(\boldsymbol{\theta}^*, K, T)$.*

The proof of the above lemma is given in Appendix F.2. Lem. 9 precisely shows a reduction of $\mathcal{I}^{clb}$ to $\mathcal{I}^{cdb}$. The claim of Lem. 9 now follows from the regret definitions of the $\mathcal{I}^{clb}$ and $\mathcal{I}^{cdb}$, precisely we can show for any fixed $T$, $2R_T^{cdb} = R_{2T}^{clb}$. Complete proof is deferred to Appendix F.1.   □

Our lower bound result now immediately follows as a implication of Thm. 10 and from the existing lower bound result of $K$-armed $d$-dimensional contextual linear bandits problem [13].

**Theorem 10** (SPM$(\boldsymbol{\theta}^*, d, 2)$: Regret Lower Bound). *For any algorithm $\mathcal{A}^{cdb}$ for the problem of linear-score based stochastic $K$-armed contextual dueling bandit of dimensional-d, there exists a sequence of $d$-dimensional context sets $\{\mathbf{x}_1^t, \ldots \mathbf{x}_K^t\}_{t=1}^T$ and a constant $\gamma > 0$ such that the regret incurred by $\mathcal{A}^{cdb}$ on $T$ rounds is at least $\Omega(\frac{\gamma}{2}\sqrt{2dT})$, i.e. $R_T(\mathcal{A}^{cdb}) \geq \frac{\gamma}{2}\sqrt{2dT}$, for any $T \geq d^2$.*

## 5   Analysis for General Subsetwise Preference Feedback (any $q \in [K]$)

We now extend our analysis to any general $q$-subsetwise feedback, where at round $t$ the learner is permitted to play a subset $\mathcal{X}_t \subseteq \mathcal{S}_t$ of size $q \geq 2$, given a fixed $q \leq K$ (formal setup in Sec. 2.1). We first analyze the regret lower bound, which, somewhat surprisingly, turns out to be independent of $q$ (Thm. 11). We also propose an algorithm following this. Proof details are given in Appendix G.

### 5.1   Regret Lower Bound

We first show that for any given $q \geq 2$, there exists a problem instance where no learner can achieve a better learning rate than $\Omega(\sqrt{dT})$. However the conclusions are much similar to the existing regret lower bounds for finite $K$ arm preference bandits for Plackett-Luce (PL) model [30, 33]. As described in Sec. 2.1, since our preference model can also be seen a special case of PL model, our results show even in the contextual framework, the learner can not attain a faster rate by playing larger subsets.

**Theorem 11** (Regret Lower Bound (Subsetwise Preferences)). *Given any $q \geq 2$ and $d > 1$, for any algorithm $\mathcal{A}$ for the problem of stochastic $K$-armed $d$-dimensional linear contextual bandit with SPM$(\boldsymbol{\theta}, d, q)$ feedback model, there exists a sequence of $d$-dimensional context sets $\{\mathbf{x}_1^t, \ldots \mathbf{x}_K^t\}_{t=1}^T$ and a choice of $\boldsymbol{\theta}$ such that the regret incurred by $\mathcal{A}$ on $T$ rounds is at least $\Omega(\sqrt{dT})$.*

### 5.2   Algorithm and Regret Guarantee

Given the above lower bound, the first thing to note is our Alg. 3, itself yields an optimal $\tilde{O}(\sqrt{dT})$ algorithm (which only makes pairwise queries per round) in case the problem allows the learner to query preferences of any subsets of size $1, 2, \ldots, q$. However, we here propose a general version of Alg. 3 which is also based on the idea of *stagewise-elimination* but can exploit subsetwise preferences for any general $q \geq 2$; it works even if the learner is restricted to play only sets of size $q$. Moreover, even though for the worst case instances, a better regret guarantee is not possible (as shown in Thm. 11), it can exploit the problem structure when there is a sufficient 'quality gap' between items.

**Main Ideas.** The main idea is to exploit the subsetwise feedback using the idea of *rank-breaking* [21] for extracting pairwise estimates from subsetwise feedback. Given these pairwise estimates, now the algorithm may proceed the same as the original Alg. 3, however instead of selecting a pair of arms, we can now select a subset of $q$ most-promising arms: First, by selecting a potential good arm and then recursively selecting the best challenger of the already selected items. The complete description is given in Alg. 4 (Appendix G.2). The challenging part, however, lies in its regret analysis which requires justifying the right concentration rates of $\boldsymbol{\theta}_t^s$, obtained from the above pairwise estimates. The detailed regret analysis is given in Appendix G.3, which finally lead to the following guarantee:

**Theorem 12** (Regret bound of Sta′D++ (Alg. 4)). *Consider any $\delta > 0$, and suppose we set the parameters of Sta′D++ (Alg. 3) as $\eta = \frac{3}{2\kappa}\sqrt{2\log\frac{3qTK}{\delta}}$, and $t_0 = 2\left(\frac{C_1\sqrt{d}+C_2\sqrt{\log(2q/\delta)}}{\lambda_{\min}(B)}\right)^2 + \frac{4\Lambda}{\lambda_{\min}(B)}$, where $\Lambda$, $\kappa$, $B$ is as defined in Lem. 4. Then with probability at least $(1-\delta)$, the $T$ round cumulative regret of Sta′D++ is at most $O\left(\frac{\sqrt{dT\log(T)}}{\kappa}\sqrt{\log\left(\frac{qTK}{\delta}\right)\log\left(\frac{Td}{\kappa}\log\frac{q}{\delta}\right)}\right)$.*

## 6 Experiments

This section gives empirical performances of our algorithms (Alg. 1 and 3) and compare them with some existing preference learning algorithms. The details of the algorithms are given below:

**Algorithms.** 1. MaxInP: Algorithm Maximum-Informative-Pair (Alg. 1 as described in Sec. 3.1). 2. Sta′D: Our proposed algorithm Stagewise-Adaptive-Duel (Alg. 1 as described in Sec. 3.2). 3 *SS*: *Self-Sparring* (independent beta priors on each arm) algorithm for multi-dueling bandits [37] 4. *RUCB*: *The Relative Upper Confidence Bound* algorithm for regret minimization in standard dueling bandits [46]. 5. *DTS*: *Dueling-Thompson Sampling* algorithm for best arm identification problem in bayesian dueling bandits [17]. In every experiment, the performances of the algorithms are measured in terms of *cumulative regret* (sec. 1), averaged across 50 runs, reported with standard deviation.

**Constructing Problem Instances.** The difficulty of the instances depends on the difference of scores of the best and second best arms, which, in the hindsight, is actually governed by the 'worst case slope' of the sigmoid function $\kappa$ (see the dependency of $\kappa$ in Thm. 3 or Thm. 5), and also by the underlying problem parameter $\boldsymbol{\theta}^* \in \mathbb{R}^d$. So we used 3 different *linear score based* problem instances based on 3 different characterizations of $\boldsymbol{\theta}^* \in \mathbb{R}^d$ (with $K$ arms and dimension $d$): **1. Easy** $h(d, K)$, **2. Extreme** $e(d, K)$, and **3. Intermediate** $m(d, K)$, by suitably adjusting the norm $\|\boldsymbol{\theta}^*\|_2$. Also in all settings, the $d$-dimensional feature vectors (of the arm set) are generated as random linear combination of each arm to be a random linear combination of the d-dimensional basis vectors (for scaling issues of the item scores, we limit each instance vector to be within ball of radius 1, i.e. $\ell_2$-norm upper bounded by 1).

**Regret vs Time.** For this experiment we fix $d = 10$ and $K = 50$. Fig. 2 shows both our algorithms MaxInP and Sta′D always outperform the rest, and their performance gets comparatively better with increasing hardness of the instances. As expected, *RUCB* performs the worst as by construction it fails to exploit the structure of underlying linear score based item preferences, due to the same reason *SS* performs poorly as well (note we implement independent armed version of the *Self-Sparring* algorithm [37] for this case, and later the *Kernelized* version for the case of non-linear item scores). On the contrary, *DTS* performs reasonably well as it designed to exploit the underlying utility structures in the pairwise-preferences.

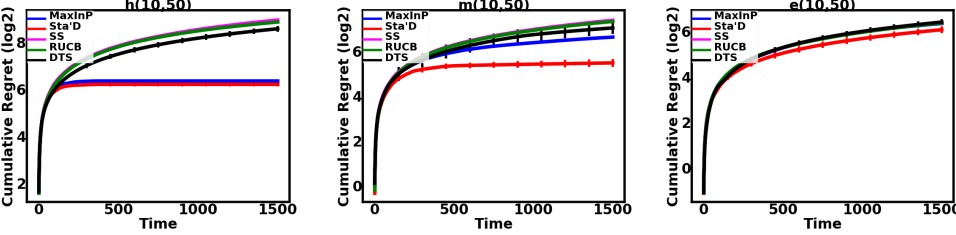

Figure 2: Average Cumulative Regret vs Time across algorithms on 3 problem instances (linear score based preferences, $d = 10, K = 50$)

**Regret vs Context-size (K).** We now compare the (averaged) final cumulative regret of each algorithm over varying context set size $(K)$ over two different problem instances. For this experiment we fix $d = 10$ and $T = 1500$. From Fig. 3 note that again our algorithms superiorly outperforms the other baselines with *DTS* performing competitively. *SS* and *RUCB* performs very badly due to the same reason as explained for Fig 2. Interesting observation to make is that the performance of both our algorithms MaxInP and Sta′D is almost independent of $K$ as also follows from their respective regret guarantees (see Thm. 3 and Thm. 5)–as long as $d$ is fixed our algorithms clearly could identify the best item irrespectively of the size $K$ of the context set, owning to their ability to exploit the underlying preference structures, unlike *SS* or *RUCB*.

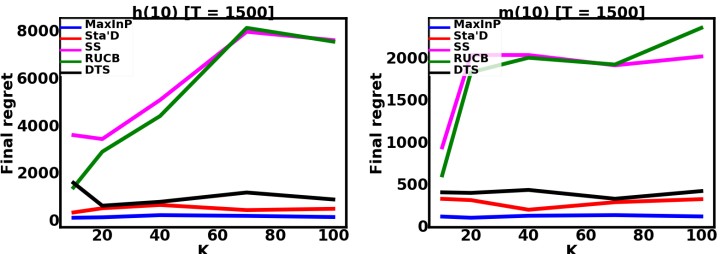

Figure 3: Final regret (averaged) vs context-set size $(K)$ across different algorithms on two different problem instances $(d = 10)$

**Regret vs Dimension (d).** For this experiment we fix $K = 80$ and $T = 1500$. From Fig. 4 shows that in general the performance of every algorithm degrades over increasing $d$. However the effect is much most severe for the *DTS* baseline compared to ours. Since *RUCB* can not exploit the underlying preference structure, its performance is mostly independent of $d$ and same goes for *SS* as well due to the same reason. The interesting observation to make is with increasing $d$, fixed $T$ and $K$, our first algorithm MaxInP indeed performs worse than Sta′D, following their theoretical regret guarantees which shows the former has a multiplicative $O(\sqrt{d})$ worse regret than latter (see Thm. 3, 5).

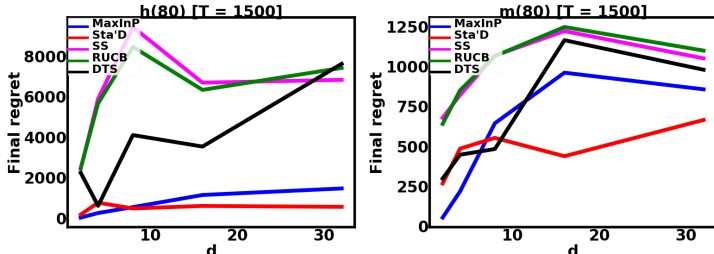

Figure 4: Final regret (averaged) vs featue dimension $(d)$ across algorithms on two different problem instances $(K = 80)$

**Non-Linear score based preferences** We finally also run some experiments to compare our regret performances on non-linear score based preferences (i.e. the score function $g(\mathbf{x})$ is not linear in $\mathbf{x}$, see Sec. 2.1 for details). We use three different score functions for the non-linear setup.

**Environments.** We use thsese 3 functions as $g(\cdot)$: **1.** *Quadratic*, **2.** *Six-Hump Camel* and 3. *Gold Stein*. *Quadratic* is the reward function $f(\mathbf{x}) = \mathbf{x}^\top H \mathbf{x} + \mathbf{x}^\top \mathbf{w} + c$, where $H \in [-1, 1]^{d \times d}$, $\mathbf{w} \in [-1, 1]^d$ and $c \in [-1, 1]$ are randomly generated. The *Six-Hump Camel* and *Gold Stein* functions are as described in [17]. For all cases, we fix $d = 3$ and $K = 50$.

**Algorithms.** We use a slightly modified version of our two algorithms (MaxInP and Sta′D) for the non-linear scores, since the GLM based parameter estimation techniques would no longer work here. But unfortunately, without suitable assumptions, we do not have an efficient way to estimate the score functions for this general setup, so instead we fit a GP to the underlying unknown score function $g(\cdot)$ based on the Laplace approximation based technique suggested in [29] (see Chap 3). For *SS* also we now used the *kernelized self-sparring* version of the algorithm [37], and for *DTS* we now fit a GP model (instead of a linear model).

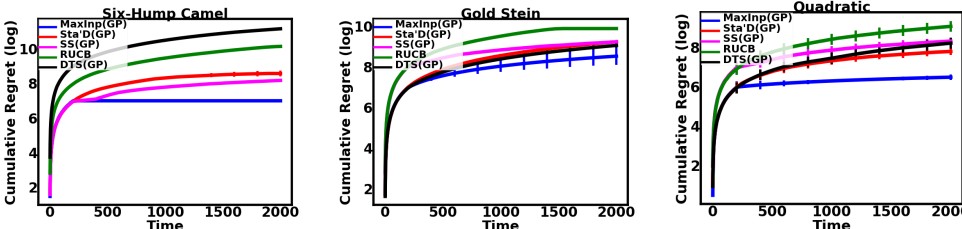

Figure 5: Avg. Cumulative Regret vs Time across algorithms on 3 problem instances (non-linear score based preferences, $d = 10, K = 50$)

Fig. 5 shows our algorithms still outperform the rest in almost all the instances. This actually implies the generality of our algorithmic ideas which applies beyond linear-scores (hence it is also worth understanding their theoretical guarantees for this general setup in future works). Moreover, unlike the previous scenarios *SS*, now starts to perform better since it could now exploit underlying preferences structures owing to the implementation of *kernelized self-sparring* [37].

## 7 Conclusion and Future Scopes

We consider the problem of regret minimization for contextual preference bandits for potentially infinite decision spaces. To the best of our knowledge, this is the first work to give optimal regret (up to logarithmic factors) $\tilde{O}(\sqrt{dT})$ algorithms along with a matching lower bound analysis. The problem of contextual preference bandits being a niche and highly practically relevant area, undoubtedly there are numerous interesting open threads to pursue along this direction: E.g. considering other link functions (probit, nested logit, etc.) based on the real-world system needs, analyzing the regret bound for adversarial preferences, or even extending preferences bandits setup to other related bandit frameworks like side information [27, 22], or feedback graphs [4, 5] etc. Analyzing instance dependent regret guarantees also remains to be an interesting future direction to see what improvements can be claimed for larger $q$ under a sufficient 'quality gap' between the items.

## Acknowledgment

Sincere thanks to Branislav Kveton and Ofer Meshi for some of the initial discussions when author was an intern at Google Research, Mountain View. Special thanks to Branislav Kveton for his insightful guidance on basics of linear-bandits and towards developing Algorithm 1.

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
