# Supplementary: Optimal Algorithms for Stochastic Contextual Preference Bandits

## A    Related Works (detailed)

The problem of regret minimization for multiarmed bandits (MAB) is very well studied in the online learning literature [7, 2, 6, 20, 25], where the learner gets to see a noisy draw of absolute reward feedback of an arm upon playing a single arm per round. However, the classical multiarmed bandits only consider finitely many arms (i.e. finite decision set) [7, 2, 25], whereas in practice, it is much more realistic to consider large decision spaces with potentially infinitely many actions, which is the reason that continuum extensions of MABs are widely studied in the literature – this includes *linear bandits* [14, 13, 1] where the true mean rewards of the arms are some linear functions of the arm features, *GLM bandits* [16, 26], where instead of linear rewards, the expected rewards of the arms follow generalized linear models (GLMs), or more generally *GP Bandits* [36] where the arms' rewards are assumed to be non-linear functions of the arm features.

On the other hand, relative feedback variants of stochastic MAB problem have also been widely studied: The most popular one being the *Dueling Bandit*, where, instead of getting a noisy feedback of the reward of the chosen arm, the learner only gets to see a noisy feedback on the pairwise preference of two arms selected by the learner. The objective is to find a high-value arm in the stochastic model and algorithmic approaches based on both upper-confidence-bounds (UCBs) [46, 23] and Thompson sampling [43] are known. There are also very few recent developments on the subsetwise extension on *Dueling Bandit* problem [38, 10, 31, 32, 30, 11]. Some of the existing work also explicitly consider the Plackett-Luce parameter estimation problem with subset-wise feedback but for offline setup only [21, 12, 19].

Surprisingly though, following the same spirit of extending MAB to continuous decision spaces (as in *linear* or *GP-bandits*), there has been very little work on the continuous extension of Dueling Bandit problem [24], that too without any theoretical performance guarantees [38, 17]. Although [24] considers the problem of dueling bandits on continuous arm set, the underlying score/reward function of each arm needs to be twice continuously differentiable, lipschitz, strongly convex as well as smooth which are very restrictive assumptions to model the preference feedback. In a recent work, [28] considers the problem of $k$-way assortment selection, where the problem is to minimize regret with respect to the set of highest revenue—again, this objective is much different than ours, which focuses on regret with respect to the single best item per iteration and hence our pairwise action set allows repeated items, unlike their setup, due to which their algorithm does not lead to sublinear regret in our case. The recent works by [38, 10, 17] did address the problem of regret minimization in continuous *Dueling Bandits*, or even the subsetwise generalization of the setting termed as *Multi-dueling bandits* with *Sparring* based [3] thompson sampling algorithm. However, none of these works analyzed any finite horizon regret guarantee of their proposed algorithms, which remains the primary objective of our work.

## B    Appendix for Sec. 2

### B.1    Derivations for Rem. 1

**Claim:** $\frac{R_T}{4e} \leq R_T^{(DB)} \leq \frac{R_T}{2}$.

*Proof.* Recall $r_T^{\text{(DB)}} = \sum_{t=1}^{T} \frac{\mathbf{P}(i_*, i_t) + \mathbf{P}(i_*, j_t) - 1}{2}$.

Note that:

$$Pr(\mathbf{x}^*, \mathbf{x}_t) - \frac{1}{2} = \frac{(e^{\mathbf{x}^{*\top}\boldsymbol{\theta}^*} - e^{\mathbf{x}_t^\top \boldsymbol{\theta}^*})}{2(e^{\mathbf{x}^{*\top}\boldsymbol{\theta}^*} + e^{\mathbf{x}_t^\top \boldsymbol{\theta}^*})} = \frac{(e^{\boldsymbol{\theta}^{*\top}(\mathbf{x}^* - \mathbf{x}_t)} - 1)}{2(e^{\boldsymbol{\theta}^{*\top}(\mathbf{x}^* - \mathbf{x}_t)} + 1)} \leq \frac{(\mathbf{x}^* - \mathbf{x}_t)^\top \boldsymbol{\theta}^*}{2}, \left[ \text{ since } \boldsymbol{\theta}^{*\top}(\mathbf{x}^* - \mathbf{x}_t) \geq 0 \right],$$

where the last inequality follows since $(e^{\boldsymbol{\theta}^{*\top}(\mathbf{x}^*-\mathbf{x}_t)} - 1) \leq \boldsymbol{\theta}^{*\top}(\mathbf{x}^* - \mathbf{x}_t)\left(\frac{1}{1 - \frac{\boldsymbol{\theta}^{*\top}(\mathbf{x}^*-\mathbf{x}_t)}{2}}\right) \leq$
$2(\boldsymbol{\theta}^{*\top}(\mathbf{x}^* - \mathbf{x}_t))$ since $\|\mathbf{x}^*\| \leq 1$ and $\|\boldsymbol{\theta}^*\| \leq 1$, then applying Cauchy-Schwartz and by the definition of $\mathbf{x}^*$ we get $\boldsymbol{\theta}^{*\top}(\mathbf{x}^* - \mathbf{x}_t) \leq \boldsymbol{\theta}^{*\top}\mathbf{x}^* \leq 1$. Moreover since $e^x - 1 > x$ for any $x > 0$, we also have:

$$Pr(\mathbf{x}^*, \mathbf{x}_t) - \frac{1}{2} = \frac{(e^{\mathbf{x}^{*\top}\boldsymbol{\theta}^*} - e^{\mathbf{x}_t^\top\boldsymbol{\theta}^*})}{2(e^{\mathbf{x}^{*\top}\boldsymbol{\theta}^*} + e^{\mathbf{x}_t^\top\boldsymbol{\theta}^*})} \geq \frac{(\mathbf{x}^* - \mathbf{x}_t)^\top\boldsymbol{\theta}^*}{4e}, \left[\text{ since } \boldsymbol{\theta}^{*\top}(\mathbf{x}^* - \mathbf{x}_t) \geq 0\right].$$

where the last inequality follows since $\|\mathbf{x}^*\| \leq 1$, $\|\mathbf{x}_t\| \leq 1$, $\|\boldsymbol{\theta}^*\| \leq 1$ and hence applying Cauchy-Schwartz both $\mathbf{x}^{*\top}\boldsymbol{\theta}^*$ and $\mathbf{x}_t^\top\boldsymbol{\theta}^* \leq 1$. Note that the same inequalities can be applied for $Pr(\mathbf{x}^*, \mathbf{y}_t) - \frac{1}{2}$ as well. Finally combining above claims and summing over $t = 1, 2, \ldots T$ we get $\frac{R_T}{4e} \leq R_T^{(DB)} \leq \frac{R_T}{2}$. $\qquad\square$

## C   Connection to GLM Bandit's Feedback Model

We start by observing the relation of our preference feedback model to that of generalized linear model (GLM) based bandits [16, 26]–precisely the feedback mechanism. The setup of *GLM bandits* generalizes the stochastic *linear bandits* problem [14, 1], where at each round $t$ the learner is supposed to play a decision point $\mathbf{x}_t$ from a set fixed decision set $\mathcal{D} \subset \mathbb{R}^d$, upon which a noisy reward feedback $f_t$ is revealed by the environment such that $f_t = \mu(\mathbf{x}_t^\top\boldsymbol{\theta}^*) + \varepsilon_t$, where $\boldsymbol{\theta}^* \in \mathbb{R}^d$ is some unknown fixed direction, $\mu : \mathbb{R} \mapsto \mathbb{R}$ is a fixed strictly increasing link function, and $\varepsilon_t$ is a zero mean $\nu$ sub-Gaussian noise for some universal constant $\nu > 0$, i.e. $\mathbf{E}\left[e^{\lambda\varepsilon_t} \mid \mathcal{H}_t\right] \leq e^{\frac{\lambda^2\nu^2}{2}}$ and $\mathbf{E}[\varepsilon_t \mid \mathcal{H}_t] = 0$ (here $\mathcal{H}_t$ denotes the sigma algebra generated by the history $\{(x_\tau, o_\tau)\}_{\tau=1}^t$ till time $t$).

The important connection now to make is that our structured dueling bandit feedback can be modeled as a GLM feedback model on the decision space of pairwise differences $\mathcal{D}' := \{(\mathbf{x} - \mathbf{y}) \mid \mathbf{x}, \mathbf{y} \in \mathcal{D}\}$, since in this case the feedback received by the learner upon playing a duel $(\mathbf{x}_t, \mathbf{y}_t)$ can be seen as: $o_t = \sigma\left((d_t)^\top\boldsymbol{\theta}^*\right) + \varepsilon_t'$ where $\varepsilon_t'$ is a 0-mean $\mathcal{H}_t$-measurable random binary noise such that

$$\varepsilon_t' = \begin{cases} 1 - \sigma\left(d_t^\top\boldsymbol{\theta}^*\right), & \text{with probability } \sigma\left(d_t^\top\boldsymbol{\theta}^*\right), \\ -\sigma\left(d_t^\top\boldsymbol{\theta}^*\right), & \text{with probability } \left(1 - \sigma\left(d_t^\top\boldsymbol{\theta}^*\right)\right), \end{cases}$$

where we denote $d_t := (\mathbf{x}_t - \mathbf{y}_t) \in \mathcal{D}'$, and it is easy to verify that $\varepsilon_t'$ is $\frac{1}{2}$ sub-Gaussian. Thus our dueling based preference feedback model can be seen as a special case of GLM bandit feedback on the decision space $\mathcal{D}'$ where the link function $\mu(\cdot)$ in our case is the sigmoid $\sigma(\cdot)$.

The above connection is crucially used in both of our proposed algorithms (Sec. 3.1 and 3.2) for estimating the unknown parameter $\boldsymbol{\theta}^*$, denoted by $\hat{\boldsymbol{\theta}}_t$, with high confidence using maximum likelihood estimation on the observed pairwise preferences $\{(\mathbf{x}_\tau, \mathbf{y}_\tau, o_\tau)\}_{\tau=1}^{t-1}$ upto time $(t-1)$, following the same technique suggested by [16, 26].

## D   Appendix for Sec. 3

### D.1   Proof of Thm. 3

**Theorem 3** (Regret bound of Maximum-Informative-Pair (Alg. 1)). *Let* $\eta = \frac{1}{2\kappa}\sqrt{\frac{d}{2}\log(1 + \frac{2T}{d}) + \log\frac{1}{\delta}}$, *where* $\kappa := \inf_{\|x-y\|\leq 2, \|\boldsymbol{\theta}^*-\hat{\boldsymbol{\theta}}\|\leq 1}\left[\sigma'\left((\mathbf{x} - \mathbf{y})^\top\hat{\boldsymbol{\theta}}\right)\right]$ *is the minimum slope of the estimated sigmoid when* $\hat{\boldsymbol{\theta}}$ *is sufficiently close to* $\boldsymbol{\theta}^*$ $\left(\sigma'(\cdot)$ *being the first order derivative of the sigmoid function* $\sigma(\cdot)\right)$. *Then given any* $\delta > 0$, *with probability at least* $(1 - 2\delta)$, *the* $T$ *round cumulative regret of Maximum-Informative-Pair satisfies:*

$$R_T \leq t_0 + \left(\frac{1}{\kappa}\sqrt{\frac{d}{2}\log\left(1 + \frac{2T}{d}\right) + \log\frac{1}{\delta}}\right)\sqrt{2dT\log\left(\frac{4t_0 + T}{d}\right)} = O\left(d\sqrt{T}\log\left(\frac{T}{d\delta}\right)\right),$$

*where we choose* $t_0 = 2 \left( \frac{C_1 \sqrt{d} + C_2 \sqrt{\log(1/\delta)}}{\lambda_{\min}(B)} \right)^2 + \frac{4}{\lambda_{\min}(B)}$, $B = \mathbf{E}_{\mathbf{x},\mathbf{y} \overset{iid}{\sim} \mathcal{P}_{\mathcal{D}}} [(\mathbf{x} - \mathbf{y})(\mathbf{x} - \mathbf{y})^\top]$ *(for some universal problem independent constants $C_1, C_2 > 0$).*

*Proof.* Our choice of $t_0$ ensures that with probability at least $(1 - \delta)$, $V_{t_0+1}$ is full rank. More precisely $\lambda_{\min}(V_{t_0+1}) \geq 1$ owing to the following standard results from random matrix theory:

**Lemma 13.** *Suppose* $\{(\mathbf{x}_1, \mathbf{y}_1), (\mathbf{x}_2, \mathbf{y}_2, \ldots, (\mathbf{x}_n, \mathbf{y}_n)\}$ *be a sequence of $n$ arm-pairs such that all* $\mathbf{x} \in \{\mathbf{x}_\tau, \mathbf{y}_\tau\}_{\tau=1}^n$ *are drawn iid from some fixed distribution $\mathcal{P}$, $\|\mathbf{x}\|_2 \leq 1$. Then for any positive constant $C > 0$, and any $\delta \in (0, 1)$, there exist two positive constants $C_1$ and $C_2$ such that if we choose* $n > 2 \left( \frac{C_1 \sqrt{d} + C_2 \sqrt{\log(1/\delta)}}{\lambda_{\min}(B)} \right)^2 + \frac{4C}{\lambda_{\min}(B)}$, *then* $Pr \left( \lambda_{\min} \left[ \sum_\tau^n (\mathbf{x}_\tau - \mathbf{y}_\tau)(\mathbf{x}_\tau - \mathbf{y}_\tau)^\top \right] \geq C \right) \geq (1 - \delta)$, *where* $B = \mathbf{E}_{\mathbf{x},\mathbf{y} \overset{iid}{\sim} \mathcal{P}} [(\mathbf{x} - \mathbf{y})(\mathbf{x} - \mathbf{y})^\top]$.

*Proof.* The result follows from the existing results of [26] (Proposition 1), which is adapted from [42] (Thm. 5.39), except we need to carefully construct the sample complexity bound considering that in our case all the iid vectors $(\mathbf{x}_t - \mathbf{y}_t) \in \mathbb{R}^d$ belong to a ball of radius 2. $\qquad \square$

We next derive the two key concentration lemmas, Lem. 1 and Lem. 2) that holds straightforwardly from the existing results of generalized linear bandits [16, 26], owing to the connection of our structured dueling bandits problem setup to that of GLM bandits.

The rest of the proof lies in expressing the regret bound in terms of the above concentration results which is possible owing to our *'most informative pair'* based arm selection strategy, as described below:

Now recall that the instantaneous regret at $t$: $r_t = \frac{(\mathbf{x}_t^* - \mathbf{x}_t)^\top \boldsymbol{\theta}^* + (\mathbf{x}_t^* - \mathbf{y}_t)^\top \boldsymbol{\theta}^*}{2}$. Then using above conditions and the by our arm selection strategy:

$$2r_t = (\mathbf{x}_t^* - \mathbf{x}_t)^\top \boldsymbol{\theta}^* + (\mathbf{x}_t^* - \mathbf{y}_t)^\top \boldsymbol{\theta}^*$$

$$= (\mathbf{x}_t^* - \mathbf{x}_t)^\top \hat{\boldsymbol{\theta}}_t + (\mathbf{x}_t^* - \mathbf{x}_t)^\top (\boldsymbol{\theta}^* - \hat{\boldsymbol{\theta}}_t) + (\mathbf{x}_t^* - \mathbf{y}_t)^\top \hat{\boldsymbol{\theta}}_t + (\mathbf{x}_t^* - \mathbf{y}_t)^\top (\boldsymbol{\theta}^* - \hat{\boldsymbol{\theta}}_t)$$

$$\overset{(1)}{\leq} \eta \|\mathbf{x}_t^* - \mathbf{x}_t\|_{V_t^{-1}} + \|\boldsymbol{\theta}^* - \hat{\boldsymbol{\theta}}_t\|_{V_t} \|\mathbf{x}_t^* - \mathbf{x}_t\|_{V_t^{-1}} + \eta \|\mathbf{x}_t^* - \mathbf{y}_t\|_{V_t^{-1}} + \|\boldsymbol{\theta}^* - \hat{\boldsymbol{\theta}}_t\|_{V_t} \|\mathbf{x}_t^* - \mathbf{y}_t\|_{V_t^{-1}}$$

$$\overset{(2)}{\leq} \eta \|\mathbf{x}_t^* - \mathbf{x}_t\|_{V_t^{-1}} + \eta \|\mathbf{x}_t^* - \mathbf{x}_t\|_{V_t^{-1}} + \eta \|\mathbf{x}_t^* - \mathbf{y}_t\|_{V_t^{-1}} + \eta \|\mathbf{x}_t^* - \mathbf{y}_t\|_{V_t^{-1}}$$

$$\overset{(3)}{\leq} \eta \|\mathbf{x}_t - \mathbf{y}_t\|_{V_t^{-1}} + \eta \|\mathbf{x}_t - \mathbf{y}_t\|_{V_t^{-1}} + \eta \|\mathbf{x}_t - \mathbf{y}_t\|_{V_t^{-1}} + \eta \|\mathbf{x}_t - \mathbf{y}_t\|_{V_t^{-1}}$$

$$= \left( \frac{2}{\kappa} \sqrt{\frac{d}{2} \log \left( 1 + \frac{2T}{d} \right)} + \log \frac{1}{\delta} \right) \|\mathbf{x}_t - \mathbf{y}_t\|_{V_t^{-1}},$$

where inequality (1) holds since since both $\mathbf{x}_t, \mathbf{y}_t \in \mathcal{C}_t$, by definition of $\mathcal{C}_t$ this implies: $(\mathbf{x}_t^* - \mathbf{x}_t)^\top \hat{\boldsymbol{\theta}}_t < \eta \|\mathbf{x}_t^* - \mathbf{x}_t\|_{V_t^{-1}}$, and $(\mathbf{x}_t^* - \mathbf{y}_t)^\top \hat{\boldsymbol{\theta}}_t < \eta \|\mathbf{x}_t^* - \mathbf{y}_t\|_{V_t^{-1}}$. Inequality (2) follows from Lem. 2, and (3) follows from the arm selection strategy. The final inequality follows by simply replacing the value of $\eta$. We now proceed to bound the cumulative regret as follows:

$$R_t = \sum_{t=1}^T r_t = \sum_{t=1}^{t_0} r_t + \sum_{t=t_0+1}^T r_t$$

$$\overset{(1)}{\leq} t_0 + \sum_{t=t_0+1}^T r_t \leq t_0 + \frac{1}{2} \sum_{t=t_0}^T \left( \frac{2}{\kappa} \sqrt{\frac{d}{2} \log \left( 1 + \frac{2T}{d} \right)} + \log \frac{1}{\delta} \right) \|\mathbf{x}_t - \mathbf{y}_t\|_{V_t^{-1}}$$

$$\overset{(2)}{\leq} t_0 + \left( \frac{1}{\kappa} \sqrt{\frac{d}{2} \log \left( 1 + \frac{2T}{d} \right)} + \log \frac{1}{\delta} \right) \sqrt{2dT \log \left( \frac{4t_0 + T}{d} \right)}$$

where the first inequality holds since $\|\mathbf{x}_t^*\| \leq 1$ and $\|\boldsymbol{\theta}^*\| \leq 1$. Thus by applying Cauchy-Schwartz and by the definition of $\mathbf{x}_t^*$ we get $\boldsymbol{\theta}^{*\top}(\mathbf{x}_t^* - \mathbf{x}) \leq \boldsymbol{\theta}^{*\top}\mathbf{x}_t^* \leq 1 \, \forall \mathbf{x} \in \mathcal{D}$–we consider the trivial bound $r_t = 1$ for the initial $t_0$ rounds of random exploration. Inequality (2) simply follows from Lem. 1, which concludes the proof. $\qquad\square$

## D.2 Proof of Lem. 1

**Lemma 1** (Self-Normalized Bound). *Suppose* $\{(\mathbf{x}_1, \mathbf{y}_1), (\mathbf{x}_2, \mathbf{y}_2), \ldots, (\mathbf{x}_t, \mathbf{y}_t)\}$ *be a sequence of arm-pair played such that all arms* $\mathbf{x} \in \{\mathbf{x}_\tau, \mathbf{y}_\tau\}_{\tau=1}^t$ *belong to the ball of unit radius. Also suppose the initial exploration length* $t_0$ *be such that* $\lambda_{\min}\left(\sum_{\tau=1}^{t_0}(\mathbf{x}_\tau - \mathbf{y}_\tau)(\mathbf{x}_\tau - \mathbf{y}_\tau)^\top\right) \geq 1$. *Then* $\forall t > t_0$,

$$\sum_{\tau=t_0+1}^t \|(\mathbf{x}_\tau - \mathbf{y}_\tau)\|_{V_{\tau+1}^{-1}} \leq \sqrt{2dt \log\left(\frac{4t_0 + t}{d}\right)}, \text{ where recall } V_{\tau+1} := \sum_{j=1}^\tau (\mathbf{x}_j - \mathbf{y}_j)(\mathbf{x}_j - \mathbf{y}_j)^\top.$$

*Proof.* As explained in Sec. C, our problem setup being a special case of GLM bandits, Lem. 1 follows directly from Lem. 2 of [26], with the additional consideration that in our case: (1). the generalized linear model is sigmoid function, (2). the subgaussianity parameter of the noise model is $\frac{1}{2}$, and (3). any arm $(\mathbf{x}_t - \mathbf{y}_t) \in \mathcal{D}' \subset R^d$ played at round $t$ belong to a ball of radius 2. $\qquad\square$

## D.3 Proof of Lem. 2

**Lemma 2** (Confidence Ellipsoid). *Suppose the initial exploration length* $t_0$ *be such that* $\lambda_{\min}\left(\sum_{\tau=1}^{t_0}(\mathbf{x}_\tau - \mathbf{y}_\tau)(\mathbf{x}_\tau - \mathbf{y}_\tau)^\top\right) \geq 1$, *and* $\kappa$ *is as defined in Thm. 3. Then for any* $\delta > 0$, *with probability at least* $(1 - \delta)$, *for all* $t > t_0$, $\|\boldsymbol{\theta}^* - \hat{\boldsymbol{\theta}}_t\|_{V_t} \leq \frac{1}{2\kappa}\sqrt{\frac{d}{2}\log\left(1 + \frac{2t}{d}\right) + \log\frac{1}{\delta}}$, *where recall* $V_{t+1} := \sum_{\tau=1}^t (\mathbf{x}_\tau - \mathbf{y}_\tau)(\mathbf{x}_\tau - \mathbf{y}_\tau)^\top$.

*Proof.* Following a similar argument as described in Lem.1, the result follows directly from Lem. 3 of [26]. $\qquad\square$

# E Appendix for Sec. 3.2

**Lemma 14** (Stagewise Sample Independence). *At any time* $t \in [T]$, *at any stage* $s \in \lfloor \log T \rfloor$, *and given an fixed realization of the played arm-pairs* $\{\mathbf{x}_\tau, \mathbf{y}_\tau\}_{\tau \in \phi^s}$, *the corresponding preference outcomes* $\{o_\tau\}_{\tau \in \phi^s}$ *are independent random variables with* $\mathbf{E}[o_\tau] = \sigma\left((\mathbf{x}_\tau - \mathbf{y}_\tau)^\top \boldsymbol{\theta}^*\right)$.

*Proof.* Note that at any stage $s \in \lfloor \log T \rfloor$ of any trial $t > t_0$, the time index $t$ is added to $\phi^s$ only if $\exists p_t^s(a_t, b_t) > \frac{1}{2^s}$. But note the value of $p_t^s(a_t, b_t)$ only depends on the other existing instances of $\phi^s$, i.e. $\{\mathbf{x}_\tau, \mathbf{y}_\tau\}_{\tau \in \phi^s}$ and not in $\{o_\tau\}_{\tau \in \phi^s}$.

Moreover, the fact that both the items $\mathbf{x} \in \{\mathbf{x}_t, \mathbf{y}_t\}$ has survived till stage $s$, means they must have passed all earlier stages $\tilde{s} < s$ which relies on their previously estimated scores $g_t^{\tilde{s}}(\mathbf{x})$ and pairwise confidence bounds $p_t^{\tilde{s}}(\mathbf{x}, j)$, $\forall j \in \mathcal{G}_t^{\tilde{s}}$—but this only depends on the observations $\cup_{\tilde{s} < s}\{\mathbf{x}_\tau, \mathbf{y}_\tau, o_\tau\}_{\tau \in \phi^{\tilde{s}}}$. And by the modelling assumption of our preference feedback, given $(\mathbf{x}_\tau, \mathbf{y}_\tau)$, $\mathbf{E}[o_\tau] = \sigma\left((\mathbf{x}_\tau - \mathbf{y}_\tau)^\top \boldsymbol{\theta}^*\right)$. Hence the claim follows. $\qquad\square$

## E.1 Pseudocode for Stagewise-Adaptive-Duel

---
**Algorithm 3 Stagewise-Adaptive-Duel (Sta′D)**

---
1: **input:** Learning rate $\eta > 0$, exploration length $t_0 > 0$
2: **init:** Select $t_0$ pairs $\{(\mathbf{x}_\tau, \mathbf{y}_\tau)\}_{\tau \in [t_0]}$, each drawn at random from $\mathcal{S}_\tau$, and observe the corresponding preference feedback $\{o_\tau\}_{\tau \in [t_0]}$
3: $S \leftarrow \lfloor \log T \rfloor$, $\phi^s \leftarrow [t_0]$, $\forall s \in [\lfloor \log T \rfloor]$
4: Set $V_{t_0+1} := \sum_{\tau=1}^{t_0} (\mathbf{x}_\tau - \mathbf{y}_\tau)(\mathbf{x}_\tau - \mathbf{y}_\tau)^\top$
5: **while** $t \leq T$ **do**
6:     $s \leftarrow 1$, $\mathcal{G}^1 \leftarrow [K]$
7:     **repeat**
8:         Compute the MLE estimate on $\phi^s$, i.e. solve for $\hat{\boldsymbol{\theta}}_t^s$ s.t.:
$$\sum_{\tau \in \phi^s} \left( o_\tau - \sigma\big((\mathbf{x}_\tau - \mathbf{y}_\tau)^\top \hat{\boldsymbol{\theta}}_t^s\big) \right)(\mathbf{x}_\tau - \mathbf{y}_\tau) = \mathbf{0}$$
9:         Set: $V_t^s = \sum_{\tau \in \phi^s} (\mathbf{x}_\tau - \mathbf{y}_\tau)(\mathbf{x}_\tau - \mathbf{y}_\tau)^\top$
10:        Compute: $g_t^s(i) = \hat{\boldsymbol{\theta}}_t^{s\top} \mathbf{x}_i^t$, $\forall i \in \mathcal{G}^s$, and $p_t^s(i,j) = \eta \|\mathbf{x}_i^t - \mathbf{x}_j^t\|_{V_t^{s-1}}$, $\forall i,j \in \mathcal{G}^s$
11:        **if** $p_t^s(i,j) \leq \frac{1}{\sqrt{T}}$, $\forall i,j \in \mathcal{G}^s$ **then**
12:           $a_t \leftarrow \arg\max_{a \in \mathcal{G}^s} g_t^s(a)$
13:           $b_t \leftarrow \arg\max_{b \in \mathcal{G}^s} \big(g_t^s(b) + p_t^s(b, a_t)\big)$
14:           Set $\mathbf{x}_t = \mathbf{x}_{a_t}^t$, $\mathbf{y}_t = \mathbf{x}_{b_t}^t$
15:        **else if** $p_t^s(i,j) \leq \frac{1}{2^s}$, $\forall i,j \in \mathcal{G}^s$ **then**
16:           Find $\mathcal{B}_t^s := \{i \in \mathcal{G}^s \mid \exists j \in \mathcal{G}^s \text{ s.t. } g_t^s(i) + \frac{1}{2^s} < g_t^s(j)\}$
17:           Update $\mathcal{G}^{s+1} \leftarrow \mathcal{G}^s \setminus \mathcal{B}^s$, $s \leftarrow s+1$
18:        **else**
19:           Choose any pair $a_t, b_t \in \mathcal{G}^s$ s.t. $p_t^s(a_t, b_t) > \frac{1}{2^s}$.
20:           Set: $\phi^s \leftarrow \phi^s \cup \{t\}$, $\mathbf{x}_t = \mathbf{x}_{a_t}^t$, $\mathbf{y}_t = \mathbf{x}_{b_t}^t$
21:        **end if**
22:     **until** a pair $(\mathbf{x}_t, \mathbf{y}_t)$ is found
23:     Play $(\mathbf{x}_t, \mathbf{y}_t)$, and update $t \leftarrow t+1$
24: **end while**

---

Using Lem. 14, one can derive sharper concentration bounds on the pairwise-arm scores as proved below:

## E.2 Proof of Lem. 4

**Lemma 4** (Sharper Concentration of Pairwise Scores). *Consider any $\delta > 0$, and suppose we set the parameters of Stagewise-Adaptive-Duel (Alg. 3) as $\eta = \frac{3}{2\kappa}\sqrt{2\log\frac{3TK}{\delta}}$, where $\kappa :=$*

$\inf_{\|\mathbf{x}-\mathbf{y}\|\leq 2, \|\boldsymbol{\theta}^* - \hat{\boldsymbol{\theta}}\| \leq 1} \left[ \sigma'\big((\mathbf{x}-\mathbf{y})^\top \hat{\boldsymbol{\theta}}\big) \right]$, *and* $t_0 = 2\left( \frac{C_1\sqrt{d} + C_2\sqrt{\log(2/\delta)}}{\lambda_{\min}(B)} \right)^2 + \frac{4\Lambda}{\lambda_{\min}(B)}$, *where* $\Lambda = \frac{8}{\kappa^4}\left(d^2 + \log\frac{3}{\delta}\right)$ *and* $B = \mathbf{E}_{\mathbf{x},\mathbf{y} \overset{iid}{\sim} \mathcal{P}_\mathcal{D}}[(\mathbf{x}-\mathbf{y})(\mathbf{x}-\mathbf{y})^\top]$ *(for some universal problem independent constants $C_1, C_2 > 0$). Then with probability at least $(1-\delta)$, for all stages $s \in \lceil \log T \rceil$ at all rounds $t > t_0$ and for all index pairs $i,j \in \mathcal{G}^s$ of round $t$: $|(\mathbf{x}_i^t - \mathbf{x}_j^t)^\top(\theta^* - \theta_t^s) \leq p_t^s(i,j)|$.*

*Proof.* The first thing to note is that due to Lem. 13, our choice of the length of initial exploration phase $t_0$ ensures that with probability at least $(1-\frac{\delta}{2})$, we have $\lambda_{\min}(V_{t_0+1}) \geq \frac{8}{\kappa^4}\left(d^2 + \log\frac{3}{\delta}\right)$.

Now recall the finite samples classical asymptotic normality of MLE estimates of GLM distributions (see Thm. 1 of [26]). Then if $\hat{\boldsymbol{\theta}}_t$ is the MLE estimate of $t$ independent random samples from any GLM model $\{Y_\tau\}_{\tau \in [t]}$ against the corresponding instance set $\{X_\tau\}_{\tau \in [t]}$, for any $\mathbf{x} \in \mathbb{R}^d$ and any $\delta > 0$, with probability at least $(1 - 3\delta)$,

$$|\mathbf{x}^\top(\hat{\boldsymbol{\theta}}_t - \boldsymbol{\theta}^*)| \leq \frac{3\gamma}{\kappa}\left( \sqrt{\log\frac{1}{\delta}} \|\mathbf{x}\|_{V_{t+1}^{-1}} \right),$$

whenever $t$ is such that $\lambda_{\min}(V_{t+1}) \geq \frac{512 M^2 \alpha^2}{\kappa^4}\left(d^2 + \log \frac{1}{\delta}\right)$, $M$ being the upper bound of the second order derivative of the GLM link function and $\gamma$ being the sub-Gaussianity parameter of the noise model, $V_{t+1} = \sum_{\tau=1}^{t} X_\tau X_\tau^\top$.

Now for specific case when the GLM link function turms out to be *the logistic / sigmoid function* $\sigma(\cdot)$ we have $M = \frac{1}{4}$, and for bernoulli noise the sub-Gaussian parameter $\gamma = \frac{1}{2}$. Thus for a GLM model with logistic link and Bernoulli noise, we now have that for any $\mathbf{x} \in \mathbb{R}^d$, with probability at least $1 - \delta$,

$$|\mathbf{x}^\top (\hat{\boldsymbol{\theta}}_t - \boldsymbol{\theta}^*)| \leq \frac{3}{2\kappa}\left(\sqrt{\log \frac{3}{\delta}} \|\mathbf{x}\|_{V_{t+1}^{-1}}\right), \tag{2}$$

whenever $\lambda_{\min}(V_{t+1}) \geq \frac{8}{\kappa^4}\left(d^2 + \log \frac{3}{\delta}\right)$.

So the coming back to our setting of algorithm Stagewise-Adaptive-Duel (Alg. 3), first note that our choice of $t_0$ already ensures that with probability at least $1 - \frac{\delta}{2}$ we have:

$$\lambda_{\min}(V_{t_0+1}) \geq \frac{8}{\kappa^4}\left(d^2 + \log \frac{3}{\delta}\right). \tag{3}$$

Then combining the result from Eqn. (2) along with the independent samples guarantee derived from Lem. 14, and owing to the connection of our preference feedback model to GLM models (as explained in Sec. C), we further have that for at any stage $s \in \lceil \log T \rceil$, at any round $t > t_0$ for any index-pair $i, j \in G^s$, denoting $\mathbf{z}_s^t(ij) = \mathbf{x}_i^t - \mathbf{x}_j^t$, with probability at least $1 - \frac{\delta}{2TK(K-1)\lceil \log T \rceil}$,

$$|(\mathbf{z}_s^t(ij))^\top (\hat{\boldsymbol{\theta}}_t - \boldsymbol{\theta}^*)| \leq \frac{3}{2\kappa}\left(\sqrt{\log \frac{6TK(K-1)\lceil \log T \rceil}{\delta}} \|\mathbf{z}_s^t(ij)\|_{V_{t+1}^{-1}}\right), \tag{4}$$

as our choice of initial exploration length $t_0$ already ensures $\lambda_{\min}(V_t^s) \geq \frac{8}{\kappa^4}\left(d^2 + \log \frac{3}{\delta}\right)$. Now taking union bound over all round $t \in T \setminus [t_0]$, all stages $s \in \lceil \log T \rceil$ and pairs $i, j \in \mathcal{G}^s$, $i \neq j$ we get that:

$$Pr\left(\forall i, j \in \mathcal{G}^s, s \in \lceil \log T \rceil \text{ of all round } t \in T \setminus [t_0], |(\mathbf{z}_s^t(ij))^\top (\hat{\boldsymbol{\theta}}_t - \boldsymbol{\theta}^*)| \right.$$

$$\left. \leq \frac{3}{2\kappa}\left(\sqrt{2 \log \frac{3TK}{\delta}} \|\mathbf{z}_s^t(ij)\|_{V_{t+1}^{-1}}\right)\right) > 1 - \frac{\delta}{2}, \tag{5}$$

upon noting for any stage $s \in \lceil \log T \rceil$, $|\mathcal{G}^s| \leq K$ and $6TK(K-1)\lceil \log T \rceil \leq (3TK)^2$. The result finally follows by taking a union bound over the two events of Eqn. (3) and (5). $\qquad\square$

### E.3 Proof of Lem. 6

**Lemma 6.** *For any $t > t_0$, suppose the pair $(\mathbf{x}_t, \mathbf{y}_t)$ is chosen at stage $s_t$, and $i_t^*$ denotes the best action of round $t$, i.e. $\mathbf{x}_{i_t^*}^t = \mathbf{x}_t^* = \arg\max_{\mathbf{x} \in \mathcal{S}_t} \mathbf{x}^\top \boldsymbol{\theta}^*$. Then for any $\delta \in (0, 1)$, with probability at least $(1-\delta)$, for all $t > t_0$: $i_t^* \in \mathcal{G}^{s_t}$ and for both $\mathbf{x} \in \{\mathbf{x}_t, \mathbf{y}_t\}$, $g(\mathbf{x}_t^*) - g(\mathbf{x}) \leq \begin{cases} \frac{2}{\sqrt{T}} & \text{if } t \in \phi^c \\ \frac{4}{2^{s_t}} & \text{otherwise} \end{cases}$.*

*Proof.* Let us consider the event $\mathcal{E} = \{\forall i, j \in \mathcal{G}^s, s \in \lceil \log T \rceil$ of all round $t \in T \setminus [t_0], |(\mathbf{x}_i^t - \mathbf{x}_j^t)^\top (\hat{\boldsymbol{\theta}}_t - \boldsymbol{\theta}^*)| \leq p_t^s(ij)\}$. The rest of the proof will assume $\mathcal{E}$ to be true which holds good with probability at least $(1 - \delta)$ as proved in Lem. 4. We will now prove the lemma breaking it into three parts:

**Part-1:** We first prove that for all $t > t_0$: $i_t^* \in \mathcal{G}^{s_t}$ following a recursive argument.

For any $t > t_0$, first note that if $s_t = 1$ (first phase) then obviously $i_t^* \in \mathcal{G}^{s_t}$ as by initialization $\mathcal{G}^{s_t} = [K]$. Now for any $s_t > 1$, if $i_t^* \notin \mathcal{G}^{s_t}$ then it must have got eliminated by some phase say $s < s_t$. But that means, at phase $s$ there was an item with index say $j \in \mathcal{G}^s \setminus \{i_t^*\}$ such that $g_t^s(j) > g_t^s(i_t^*) + \frac{1}{2^s}$.

With slight abuse of notation, let us denote for any index $i \in \mathcal{G}^s$ its true score as $g_t^*(i) := g(\mathbf{x}_i^t) = \mathbf{x}_i^{t\top}\theta^*$. Recall that $g_t^*(i) = g(\mathbf{x}_i^t) = \mathbf{x}_i^{t\top}\theta^*$, and $g_t^s(i) = \mathbf{x}_i^{t\top}\theta_t^s$. Let us also denote for any index pair $i, j \in \mathcal{G}^s$, their estimated score difference $d_t^s(i,j) := g_t^s(i) - g_t^s(j)$, and true pairwise score difference $d_t^*(i,j) := g_t^*(i) - g_t^*(j)$. So by definition $d_t^*(i_t^*, i) > 0, \forall i \in \mathcal{S}_t \setminus \{i_t^*\}$. So in particular $d_t^*(i_t^*, j) > 0$ as well.

But since both $i_t^*$ and $j$ have passed stage $s$ and we assume the event $\mathcal{E}$ to be true, from Lem. 4 we have that $|d_t^s(i_t^*, j) - d_t^*(i_t^*, j)| \leq p_t^s(i_t^*, j) \leq \frac{1}{2^s}$. But this further implies

$$d_t^s(i_t^*, j) \geq d_t^*(i_t^*, j) - \frac{1}{2^s} > -\frac{1}{2^s} \implies g_t^s(i_t^*) \geq g_t^s(j) - \frac{1}{2^s},$$

which gives a contradiction as for $i_t^*$ to get eliminated at stage $s$, we earlier assumed $g_t^s(j) > g_t^s(i_t^*) + \frac{1}{2^s}$. So $i_t^*$ must be present at stage $s_t$.

**Part-2:** We now prove that for both $\mathbf{x} \in \{\mathbf{x}_t, \mathbf{y}_t\}$, $g(\mathbf{x}_t^*) - g(\mathbf{x}) \leq \frac{2}{\sqrt{T}}$, if $t \in \phi^c$.

Recall $\mathbf{x}_t = \mathbf{x}_{a_t}^t$ and $\mathbf{y}_t = \mathbf{x}_{b_t}^t$. We would only consider the cases $a_t \neq i_t^*$ and $b_t \neq i_t^*$, as the claim is trivially true otherwise.

First let us analyse the case for $a_t \neq i_t^*$, by our arm selection strategy this means $d_t^s(a_t, i_t^*) > 0$ since both $i_t^*$ and $a_t$ are present at $s_t$. Also as $t \in \phi^c$, and we assume the event $\mathcal{E}$ to be true, from Lem. 4 we have $|d_t^s(i_t^*, a_t) - d_t^*(i_t^*, a_t)| \leq p_t^s(i_t^*, a_t) \leq \frac{1}{\sqrt{T}}$. But this further implies

$$d_t^*(a_t, i_t^*) \geq d_t^s(a_t, i_t^*) - \frac{1}{\sqrt{T}} > -\frac{1}{\sqrt{T}} \implies g_t^s(a_t) \geq g_t^s(i_t^*) - \frac{1}{\sqrt{T}},$$

Now if $a_t = b_t$, then the claim follows from the earlier bound itself. Assuming $a_t \neq b_t$ and $b_t \neq i_t^*$, once again by our arm selection strategy this implies $g_t^s(b_t) + p_t^s(b_t, i_t^*) > g_t^s(i_t^*) \implies g_t^s(b_t) > g_t^s(i_t^*) - \frac{1}{\sqrt{T}}$. Also since $t \in \phi^c$, and we assume the event $\mathcal{E}$ to be true, from Lem. 4 we have $|d_t^s(i_t^*, a_t) - d_t^*(i_t^*, b_t)| \leq p_t^s(i_t^*, b_t) \leq \frac{1}{\sqrt{T}}$, which further implies

$$d_t^*(b_t, i_t^*) \geq d_t^s(b_t, i_t^*) - \frac{1}{\sqrt{T}} > -\frac{2}{\sqrt{T}} \implies g_t^s(b_t) \geq g_t^s(i_t^*) - \frac{2}{\sqrt{T}}.$$

This validates the claim of this part.

**Part-3:** Finally in this part we show that for both $\mathbf{x} \in \{\mathbf{x}_t, \mathbf{y}_t\}$, $g(\mathbf{x}_t^*) - g(\mathbf{x}) \leq \frac{8}{2^{s_t}}$, if $t \in [T] \setminus \phi^c$

Assuming any stage $s_t \in \lfloor \log T \rfloor$, if $\mathbf{x}_t$ and $\mathbf{y}_t$ has survived till $s_t$ this means they were not eliminated by $\mathbf{x}_t^*$ at any stage $s < s_t$, as by the claim of **Part-1** $\mathbf{x}_t^*$ survives till stage $s_t$ as well.

Let us first prove the claim for $\mathbf{x}_t = \mathbf{x}_{a_t}^t$. Since its corresponding index $a_t$ did not get eliminated at stage $s_t - 1$ this implies $d_t^s(a_t, i_t^*) > \frac{1}{2^{s_t-1}}$. Moreover as we assume the event $\mathcal{E}$ to be true, from Lem. 4 we have $|d_t^s(i_t^*, a_t) - d_t^*(i_t^*, b_t)| \leq \frac{2}{2^{s_t}}$, which further implies

$$d_t^*(a_t, i_t^*) \geq d_t^s(a_t, i_t^*) - \frac{2}{2^{s_t}} \implies g_t^*(a_t) \geq g_t^*(i_t^*) - \frac{4}{2^{s_t}},$$

which proves the claim for $\mathbf{x}_t$. The same claim for $\mathbf{y}_t$ can be proved as well by simply following the exact same chain of arguments shown for $\mathbf{x}_t$. $\qquad\square$

### E.4 Proof of Lem. 7

**Lemma 7.** *At any stage $s \in \lfloor \log T \rfloor$, with probability at least $(1-\delta)$, $\sqrt{|\phi^s|} \leq \eta 2^s \sqrt{2d \log\left(\frac{4t_0 T}{d}\right)}$.*

*Proof.* Firstly note that due to Lem. 1, at any state $s \in \lfloor \log T \rfloor$ of round $T$

$$\sum_{\tau \in \phi^s} \|(\mathbf{x}_\tau - \mathbf{y}_\tau)\|_{(V_T^s)^{-1}} \leq \sqrt{2d|\phi^s|\log\left(\frac{4t_0 + |\phi^s|}{d}\right)} \leq \sqrt{2d|\phi^s|\log\left(\frac{4Tt_0}{d}\right)},$$

since our choice of $t_0$ already ensures $\lambda_{\min}(V_T^s) \geq 1$ (noting by definition $\kappa < 1$), the second inequality follows from the fact that by definition $t_0, |\phi^s| \geq 1$ (claim holds trivially if $\phi^s = \emptyset$) and also $|\phi^s| \leq T$.

Recalling that $a_t, b_t$ respectively denotes the index of the played pair $\mathbf{x}_t, \mathbf{y}_t$ at any round $t > t_0$, above further implies

$$\sum_{\tau \in \phi^s} p_\tau^s(a_\tau, b_\tau) \leq \eta \sqrt{2d|\phi^s|\log\left(\frac{4Tt_0}{d}\right)}. \tag{6}$$

But on the other hand, by the construction of sets $\phi^s$, we have: $\sum_{\tau \in \phi^s} p_\tau^s(a_\tau, b_\tau) \geq \frac{|\phi^s|}{2^s}$.

Then combining above with Eqn. (6) we have: $\frac{|\phi^s|}{2^s} \leq \sum_{\tau \in \phi^s} p_\tau^s(a_\tau, b_\tau) \leq \eta \sqrt{2d|\phi^s|\log\left(\frac{4Tt_0}{d}\right)}$,

which finally implies $\sqrt{\phi^s} \leq \eta 2^s \sqrt{2d\log\left(\frac{4Tt_0}{d}\right)}$, and the claim follows. $\qquad\square$

### E.5 Proof of Thm. 5

**Theorem 5** (Regret bound of Stagewise-Adaptive-Duel (Alg. 3)). *Consider we set $t_0$ and $\eta$ as per Lem. 4. Then for any $\delta > 0$, with probability at least $(1 - \delta)$, the regret of Alg. 3 can be bounded as:*

$$R_T \leq t_0 + 4\eta\sqrt{2d\log\left(\frac{4t_0T}{d}\right)}\sqrt{T\log T} + 2\sqrt{T} = O\left(\frac{\sqrt{dT\log T}}{\kappa}\sqrt{\log\left(\frac{TK}{\delta}\right)\log\left(\frac{Td}{\kappa}\log\frac{1}{\delta}\right)}\right)$$

*Proof.* Suppose we denote by $\phi^c := \{t \in [T] \setminus [t_0] \mid t \notin \cup_{s=1}^{\lfloor \log T \rfloor} \phi^s\}$ the set of all good time intervals where all the index pairs $p_t^s(i,j)$ are estimated within the confidence accuracy $\frac{1}{\sqrt{T}}$. The proof crucially relies on the concentration bound of Lem. 4, from which we first derive Lem. 6. Further owing to Lem. 1 and due to the construction of our *'stagewise-good item pairs'* we also derive another main result: Lem. 7.

The final regret bound now follows clubbing the results of Lem. 6 and 7 as given below:

$$R_t = \sum_{t=1}^T r_t = \sum_{t=1}^{t_0} r_t + \sum_{s=1}^{\lfloor \log T \rfloor}\sum_{t \in \phi^s} r_t + \sum_{t \in \phi^c} r_t$$

$$\overset{(a)}{\leq} t_0 + \sum_{s=1}^{\lfloor \log T \rfloor} |\phi^s|\frac{4}{2^s} + |\phi^c|\frac{2}{\sqrt{T}}$$

$$\overset{(b)}{\leq} t_0 + 4\sum_{s=1}^{\lfloor \log T \rfloor}\frac{2^s\eta\sqrt{2d|\phi^s|}}{2^s}\sqrt{\log\left(\frac{4t_0T}{d}\right)} + 2\sqrt{T}$$

$$\leq t_0 + 4\eta\sqrt{2d\log\left(\frac{4t_0T}{d}\right)}\sum_{s=1}^{\lfloor \log T \rfloor}\sqrt{|\phi^s|} + 2\sqrt{T}$$

$$\overset{(c)}{\leq} t_0 + 4\eta\sqrt{2d\log\left(\frac{4t_0T}{d}\right)}\sqrt{T\log T} + 2\sqrt{T}$$

$$= O\left(\frac{\sqrt{dT \log T}}{\kappa} \sqrt{\log\left(\frac{TK}{\delta}\right) \log\left(\frac{Td}{\kappa} \log \frac{1}{\delta}\right)}\right)$$

where recall that $\phi^c := \{t \in [T] \setminus [t_0] \mid t \notin \cup_{s=1}^{\lfloor \log T \rfloor} \phi^s\}$. We consider the trivial bound of $r_t = 1$ for the initial $t_0$ rounds. Note that here the inequality (a) follows from Lem. 6, (b) from Lem. 7 and since $\phi^c \leq T$. Inequality (c) uses Cauchy-Schwartz along with the fact that $\cup_{s=1}^{\lfloor \log T \rfloor} \phi^s \leq T$. Finally the order of the regret bound follows by considering our particular choice of $\eta$, $t_0$ and rearranging the terms. $\qquad \square$

# F   Appendix for Sec. 4

## F.1   Proof of Lem. 8

**Lemma 8** (Reducing $\mathcal{I}^{clb}$ with *Gumbel noise* to $\mathcal{I}^{cdb}$)**.** *There exists a reduction from the $\mathcal{I}^{clb}$ problem (under Gumbel noise, i.e. $\varepsilon_t \overset{iid}{\sim} Gumbel(0,1)$) to $\mathcal{I}^{cdb}$ which preserves the expected regret.*

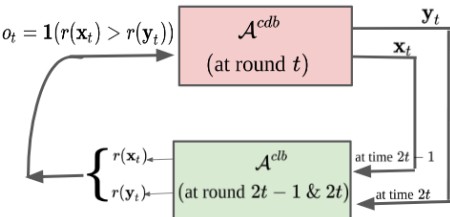

Figure 6: Pictorial demonstration of the reduction: Reducing $\mathcal{I}^{clb}$ to $\mathcal{I}^{cdb}$

*Proof.* Suppose we have a blackbox algorithm for the instance of $\mathcal{I}^{cdb}$ problem, say $\mathcal{A}^{cdb}$. To prove the claim, our goal is to show that this can be used to solve the $\mathcal{I}^{clb}$ problem where the underlying stochastic noise, $\epsilon_t$ at round $t$, is generated from a Gumbel$(0,1)$ distribution [39, 9]: Precisely we can construct an algorithm for $\mathcal{I}^{clb}(\boldsymbol{\theta}^*, K, T)$ (say $\mathcal{A}^{clb}$) using $\mathcal{A}^{cdb}$ as shown in Alg. 2.

The reduction now follows from Lem. 9 which establish the first half of the claim as it precisely shows a reduction of $\mathcal{I}^{clb}$ to $\mathcal{I}^{cdb}$. The second half of the claim is easy to follow from the corresponding regret definitions of the $\mathcal{I}^{clb}$ and $\mathcal{I}^{cdb}$ problem, Eqn. (4) and (1) respectively: Precisely owing to the reduction on Lem. 9, for any fixed $T$, $2R_T^{cdb} = R_{2T}^{clb}$. $\qquad \square$

## F.2   Proof of Lem. 9

**Lemma 9.** *If $\mathcal{A}^{clb}$ rums on a problem instance $\mathcal{I}^{clb}(\boldsymbol{\theta}^*, K, 2T)$ with Gumbel$(0,1)$ noise, then the internally the algorithm $\mathcal{A}^{cdb}$ runs on a problem instance of $\mathcal{I}^{cdb}(\boldsymbol{\theta}^*, K, T)$.*

*Proof.* Firstly it is easy to note from the construction of $\mathcal{A}^{clb}$ that one round of $\mathcal{A}^{cdb}$, say round $t \in \lceil \frac{T}{2} \rceil$, goes in two consecutive rounds of $\mathcal{A}^{clb}$, round $2t-1$ and $2t$ of $\mathcal{A}^{cdb}$.

We now show the main claim that by construction of $\mathcal{A}^{clb}$, the internal world of $\mathcal{A}^{cdb}$ indeed receives feedback from an instance of $\mathcal{I}^{cdb}(\boldsymbol{\theta}^*, K, T)$: Precisely, recalling our feedback model for any problem instance of $\mathcal{I}^{cdb}(\boldsymbol{\theta}^*, K, T)$ from Sec. 2.1, we want to establish the following claim:

**Claim:** At any round $t \in \lfloor \frac{T}{2} \rfloor$ of $\mathcal{A}^{cdb}$, $o_t = \mathbf{1}(\mathbf{x}_t \text{ preferred over } \mathbf{y}_t) \sim Ber\left(\sigma\left((\mathbf{x}_t - \mathbf{y}_t)^\top \boldsymbol{\theta}^*\right)\right)$.

Towards this note that by construction of $\mathcal{A}^{cdb}$ we have $o_t = \mathbf{1}(r(\mathbf{x}_t) > r(\mathbf{y}_t))$. Now by the setting of any problem instance $\mathcal{I}^{cdb}(\boldsymbol{\theta}^*, K, T)$ with iid Gumbel$(0,1)$ noise, note that given any $\mathbf{x} \in \mathbb{R}^d$, $r(\mathbf{x}) \sim$ Gumbel$(\mathbf{x}^\top \boldsymbol{\theta}^*, 1)$ [40]. But then given arm pair $\mathbf{x}_t$ and $\mathbf{y}_t$ and by defining $Z_t = \max(r(\mathbf{x}_t), r(\mathbf{y}_t))$, by the property of max of two independent Gumbel distributions [9, 34]:

$$Pr(Z_t = \mathbf{x}_t \mid \{\mathbf{x}_t, \mathbf{y}_t\}) = \frac{e^{\mathbf{x}_t^\top \boldsymbol{\theta}^*}}{e^{\mathbf{x}_t^\top \boldsymbol{\theta}^*} + e^{\mathbf{y}_t^\top \boldsymbol{\theta}^*}} = \frac{1}{1 + e^{(\mathbf{x}_t - \mathbf{y}_t)^\top \boldsymbol{\theta}^*}} = \sigma((\mathbf{x}_t - \mathbf{y}_t)^\top \boldsymbol{\theta}^*).$$

The result now follows noting $o_t = 1$, if $Z_t = \mathbf{x}_t$ and $o_t = 0$, if $Z_t = \mathbf{y}_t$, implying $o_t \sim \text{Ber}\big(\sigma\big(\mathbf{x}_t - \mathbf{y}_t\big)^\top \boldsymbol{\theta}^*\big)$. $\qquad\square$

### F.3 Proof of Thm. 10

**Theorem 10** (SPM($\boldsymbol{\theta}^*, d, 2$): Regret Lower Bound). *For any algorithm $\mathcal{A}^{cdb}$ for the problem of linear-score based stochastic $K$-armed contextual dueling bandit of dimensional-d, there exists a sequence of $d$-dimensional context sets $\{\mathbf{x}_1^t, \ldots \mathbf{x}_K^t\}_{t=1}^T$ and a constant $\gamma > 0$ such that the regret incurred by $\mathcal{A}^{cdb}$ on $T$ rounds is at least $\Omega(\frac{\gamma}{2}\sqrt{2dT})$, i.e. $R_T(\mathcal{A}^{cdb}) \geq \frac{\gamma}{2}\sqrt{2dT}$, for any $T \geq d^2$.*

*Proof.* The proof immediately follows from the known regret lower bound for of $K$-armed $d$-dimensional contextual linear bandits problem (see Thm. 2 of [13]), and from the fact that for any $T$, $2R_T^{cdb} = R_{2T}^{clb}$ as we proved in Lem. 8: This is because any smaller regret for $\mathcal{A}^{cdb}$ would violate the best achievable regret bound of $\mathcal{A}^{clb}$ which is a logical contradiction as this would imply $R_{2T}^{clb} = 2R_T^{cdb} < \gamma\sqrt{2dT}$. So it must be the case that $R_T^{cdb} \geq \frac{\gamma}{2}\sqrt{2dT}$. $\qquad\square$

## G  Appendix to Sec. 5

### G.1  Proof of Thm. 11

**Theorem 11** (Regret Lower Bound (Subsetwise Preferences)). *Given any $q \geq 2$ and $d > 1$, for any algorithm $\mathcal{A}$ for the problem of stochastic $K$-armed $d$-dimensional linear contextual bandit with SPM($\boldsymbol{\theta}, d, q$) feedback model, there exists a sequence of $d$-dimensional context sets $\{\mathbf{x}_1^t, \ldots \mathbf{x}_K^t\}_{t=1}^T$ and a choice of $\boldsymbol{\theta}$ such that the regret incurred by $\mathcal{A}$ on $T$ rounds is at least $\Omega(\sqrt{dT})$.*

**Remark 2.** *The interesting part here to note is that when $q = 2$ (the dueling bandit/pairwise preference case), we already know the lower bound is $\Omega(\sqrt{dT})$ from (Thm. 10). So whatever is possible to achieve for the dueling feedback is also achievable with the q-subsetwise feedback simply using $q = 2$. The question really to ask here is if its possible to achieve an faster learning rate (smaller regret) with general q-subsetwise queries. However, our lower bound result clearly shows the impossibility of any such hope.*

*Proof.* The proof relies on constructing a 'hard enough' problem instance for the learning framework and showing no algorithm can achieve a smaller rate of regret on that instance than the claimed lower bounds. We assume the decision space $\mathcal{D}$ to be the set of standard basis vectors: $\mathcal{D} = \{\mathbf{e}_1, \ldots \mathbf{e}_d\}$, and at each round $t$, the learner receives the set of context vectors $\mathcal{S}_t = \mathcal{D}$. Now let us construct $d + 1$ problem instances $\mathcal{I}^1, \mathcal{I}^2, \ldots, \mathcal{I}^d$ and $\mathcal{I}^0$, where each instance is uniquely identified by its underlying score vector $\boldsymbol{\theta}$ as defined below:

**Base Problem Instance** ($\mathcal{I}^0$): $\boldsymbol{\theta}^0(i) = 0.5, \ \forall i \in [d]$. Now let us consider $d$ alternative problem instances $\mathcal{I}^m \ \forall m \in [d]$:

**Problem Instance-$m$** ($\mathcal{I}^m$): For all $t \in [T]$, $\boldsymbol{\theta}^m(i) = 0.5, \ \forall i \in [d] \setminus \{m\}$, $\boldsymbol{\theta}^m(m) = 0.5 + \epsilon$, for some $\epsilon \in (0, 0.2]$.

Clearly, for instance $\mathcal{I}^m$, the unique 'best' item is $i_*^m := m$, and rest of all the $d - 1$ items are 'bad' playing which at any round $t \in [T]$ yields a regret of $\epsilon/|\mathcal{X}_t|$.

Let $N_T(S) := \mathbf{E}[\sum_{t=1}^T \mathbf{1}(S = \mathcal{X}_t)]$, denotes the expected number of times $\mathcal{A}$ pulls the subset $S \subseteq \mathcal{D}$ in $T$ rounds. One key remark before proceeding to the main analysis is: We consider *only the class of all deterministic algorithms*, i.e. where $\mathcal{X}_t$ is a deterministic function of the past history $\mathcal{H}_{t-1} := \{\mathcal{S}_1, \mathcal{X}_1, i_1, \ldots \mathcal{S}_{t-1}, \mathcal{X}_{t-1}, i_{t-1}, \mathcal{S}_t\}$. Note this is without loss of generality, since any randomized strategy can be seen as a randomization over deterministic querying strategies. Thus, a lower bound which holds uniformly for any deterministic class of algorithms, would also hold over a randomized class of algorithms.

Note since for any instance $\mathcal{I}^m$, $m \in [d]$, the regret in $T$ round can be lower bounded as:

$$\mathbf{E}_{\mathcal{I}^m}[R_T(\mathcal{A})] \geq \sum_{t=1}^T \left( \mathbf{E}_{\mathcal{I}^m}\big[\mathbf{1}(\mathcal{X}_t \cap [d] \setminus \{m\} \neq \emptyset)\big] \frac{\epsilon(|\mathcal{X}_t| - 1)}{|\mathcal{X}_t|} \right)$$

$$\geq \sum_{t=1}^{T} \frac{\epsilon}{2}\left(T - \mathbf{E}_{\mathcal{I}^m}\left[\mathbf{1}(\mathcal{X}_t = \{i_*^m\})\right]\right) = \frac{\epsilon}{2}\left(T - \mathbf{E}_{\mathcal{I}^m}[N_T(i_*^m)]\right).$$

Then taking average over $\mathcal{I}^m$s for all $m \in [d]$:

$$\mathbf{E}[R_T(\mathcal{A})] = \sum_{m\in[d]} \frac{\mathbf{E}_{\mathcal{I}^m}[R_T(\mathcal{A})]}{d} \geq \frac{\epsilon}{2}\left(T - \frac{\sum_{m\in[d]}\mathbf{E}_{\mathcal{I}^m}[N_T(m)]}{d}\right) \tag{7}$$

since $i_*^m = m$. Now note that:

$$\mathbf{E}_{\mathcal{I}^m}[N_T(m)] - \mathbf{E}_{\mathcal{I}^0}[N_T(m)]$$
$$= \sum_{t=1}^{T} \left(Pr_{\mathcal{I}^m}(\mathcal{X}_t = \{m\}) - Pr_{\mathcal{I}^0}(\mathcal{X}_t = \{m\})\right) \leq T.D_{TV}(\mathcal{I}^0, \mathcal{I}^m), \tag{8}$$

where $D_{TV}(\mathcal{I}^0, \mathcal{I}^m)$ denotes the total variation distance between the probability distribution of $\mathcal{I}^0$ and $\mathcal{I}^m$ with respect to history $\mathcal{H}_T$, i.e. $D_{TV}(\mathcal{I}^0, \mathcal{I}^m) := \sup_{\mathcal{E} \in \mathcal{H}_T} |Pr_{\mathcal{I}^0}(\mathcal{E}) - Pr_{\mathcal{I}^m}(\mathcal{E})|$.

Further using Pinksker's inequality we have

$$D_{TV}(\mathcal{I}^0, \mathcal{I}^m) \leq \sqrt{\frac{1}{2}D_{KL}(\mathcal{I}^0, \mathcal{I}^m)}, \tag{9}$$

where $D_{KL}(\mathcal{I}^0, \mathcal{I}^m)$ denotes the KL-divergence between the probability distribution induced on the observed history $\mathcal{H}_T$ by the problem instance $\mathcal{I}^0$ and $\mathcal{I}^m$. Thus averaging over $\mathcal{I}^m$s for all $m \in [d]$:

$$\frac{\sum_{m\in[d]}\mathbf{E}_{\mathcal{I}^m}[N_T(m)]}{d} \leq \frac{\sum_{m\in[d]}\left(\mathbf{E}_{\mathcal{I}^0}[N_T(m)] + T.D_{TV}(\mathcal{I}^0, \mathcal{I}^m)\right)}{d}$$
$$= \frac{\sum_{m\in[d]}\mathbf{E}_{\mathcal{I}^0}[N_T(m)]}{d} + T\sum_{m\in[d]}\frac{1}{d}\left(\sqrt{\frac{1}{2}D_{KL}(\mathcal{I}^0, \mathcal{I}^m)}\right)$$
$$= \frac{\sum_{m\in[d]}\mathbf{E}_{\mathcal{I}^0}[N_T(m)]}{d} + T\sqrt{\left(\frac{1}{2d}\sum_{m\in[d]}D_{KL}(\mathcal{I}^0, \mathcal{I}^m)\right)} \tag{10}$$

With slight abuse of notation, by denoting $\mathcal{I}_t^0 := Pr_{\mathcal{I}^0}\left(\mathbf{P}_t(\mathcal{X}_t) \mid \mathcal{H}_{t-1}\right)$ and $\mathcal{I}_t^m := Pr_{\mathcal{I}^m}\left(\mathbf{P}_t(\mathcal{X}_t) \mid \mathcal{H}_{t-1}\right)$, we note that:

$$D_{KL}(\mathcal{I}_t^0, \mathcal{I}_t^m) \sim \begin{cases} KL\left(\text{Mult}(\boldsymbol{\theta}^0, \mathcal{X}_t), \text{Mult}(\boldsymbol{\theta}^m, \mathcal{X}_t)\right), & \text{if } m \in \mathcal{X}_t \text{ and } |\mathcal{X}_t| > 1 \\ 0, & \text{otherwise} \end{cases},$$

where $\text{Mult}(\boldsymbol{\theta}^0, \mathcal{X}_t)$ and $\text{Mult}(\boldsymbol{\theta}^m, \mathcal{X}_t)$ respectively denote the multinomial distributions $(Pr(\cdot \mid \mathcal{X}_t, \boldsymbol{\theta}))$ induced by the instances $\mathcal{I}^0$ and $\mathcal{I}^m$ on the elements of $\mathcal{X}_t$. Let us also denote by $S^{(m)} := \{S \subseteq \mathcal{D} \mid m \in S, |S| > 1\}$.

So using chain rule of KL-divergence we get:

$$D_{KL}(\mathcal{I}^0, \mathcal{I}^m) = \sum_{t=1}^{T} D_{KL}(\mathcal{I}_t^0, \mathcal{I}_t^m) = \sum_{t=1}^{T}\sum_{S\in S^{(m)}} Pr_{\mathcal{I}^0}(\mathcal{X}_t = S)KL\left(\text{Mult}(\boldsymbol{\theta}^0, \mathcal{X}_t), \text{Mult}(\boldsymbol{\theta}^m, \mathcal{X}_t)\right)$$
$$\leq \sum_{t=1}^{T}\sum_{S\in S^{(m)}} Pr_{\mathcal{I}^0}(\mathcal{X}_t = S)\frac{180\epsilon^2}{|\mathcal{X}_t|} \leq 180\epsilon^2 \sum_{t=1}^{T}\sum_{S\in S^{(m)}}\frac{\mathbf{E}_{\mathcal{I}^0}[N_T(S)]}{|S|},$$

where the last inequality follows by noting for any set $S \subseteq \mathcal{D}$, $KL\left(\text{Mult}(\boldsymbol{\theta}^0, S), \text{Mult}(\boldsymbol{\theta}^m, S)\right) \leq 180\frac{\epsilon^2}{|S|}$ for any $\epsilon \in (0, 0.2]$. Further averaging over $\mathcal{I}^m$s for all $m \in [d]$:

$$\frac{\sum_{m\in[d]} D_{KL}(\mathcal{I}^0, \mathcal{I}^m)}{d} \leq \frac{180\epsilon^2 \sum_{m\in[d]} \sum_{S\in S^{(m)}}^{K} \mathbf{E}_{\mathcal{I}^0}[N_T(S)]}{d} \leq \frac{180\epsilon^2 T}{d}.$$

Now combining above with Eqn. (7) and 10 we get:

$$\mathbf{E}[R_T(\mathcal{A})] \geq \frac{\epsilon}{2}\left(T - \left(\frac{\sum_{m\in[d]} \mathbf{E}_{\mathcal{I}^0}[N_T(m)]}{d} + T\sqrt{\left(\frac{1}{2d}\sum_{m\in[d]} D_{KL}(\mathcal{I}^0, \mathcal{I}^m)\right)}\right)\right)$$

$$\geq \frac{\epsilon}{2}\left(T - \left(\frac{T}{d} + \epsilon T\sqrt{\left(\frac{90T}{d}\right)}\right)\right) \overset{(a)}{\geq} \frac{\epsilon}{2}\left(T - \left(\frac{T}{2} + \frac{T}{4}\right)\right) = \frac{\epsilon}{2}\left(T - \frac{3T}{4}\right) = \frac{\sqrt{dT}}{32\sqrt{90}},$$

where $(a)$ holds by setting $\epsilon = \frac{1}{4}\sqrt{\frac{d}{90T}}$ and since $d \geq 2$, and the last equality follows for the above choice of $\epsilon$, concluding the proof. □

### G.2 Pseudocode for Sta′D++ (Alg. 4)

Given the above lower bound, we see that the learner having a provision to play $q$-subsetwise queries does not help in faster learning (small regret bound), the question we started asking in Rem. 2. It is hence easy to see that our Alg. 3, itself yields an optimal $\tilde{O}(\sqrt{dT})$ algorithm using $q = 2$ at every rounds (i.e. only making pairwise queries per round). However, we here propose a general version of Alg. 3 which is also based on the idea of *stagewise-elimination* but can exploit subsetwise preferences for any general $q \geq 2$. So even though for the worst case instances a better regret guarantee is not possible (as shown in Lem. 11), the hope is it would be able to exploit the problem structure when there is sufficient 'quality-gap' (in terms of $\boldsymbol{\theta}$) between best vs the rest of the items. Analyzing such instance dependent guarantees for the problem setup would definitely be interesting but beyond the scope of this work. Following (Alg. 4) describes our algorithm for general $q$-subsetwise preferences and the corresponding worst case (i.e. instance-independent) regret analysis (Thm. 12).

**Main Ideas.** The main idea is to exploit the subsetwise feedback using the idea of *rank-breaking* [21, 35] for extracting pairwise estimates from multiwise (subsetwise) preference information. Formally, at any round $t$ if we play the set $\mathcal{X}_t$ and observe the winner $i_t$, *rank-breaking* suggests as if to treat item $i_t$ has beaten all the rest of the items in $\mathcal{X}_t$ in a pairwise duel, resulting in $|\mathcal{X}_t| - 1$ pairwise preferences of the form $(i_t \succ j)$, $\forall j \in \mathcal{X}_t \setminus \{i_t\}$. Upon extracting these pairwise duels, we can similarly get estimate the $\hat{\boldsymbol{\theta}}_{t+1}$ by running a MLE on these extracted pairwise preferences up to round $t$. Given the $\hat{\boldsymbol{\theta}}$ estimate, the algorithm then proceed same as the original Alg. 3. However instead of selecting a pair of arms, we can now select a subset of $q$ 'most-promising arms', by first selecting a potential good arm and then recursively selecting the best challenger of the already selected items using a *recursive max-min subset selection rule* as used in [33]. Details is given in Alg. 4. The challenging part however lies in its regret analysis which requires to justify the right concentration rates of $\boldsymbol{\theta}_t^s$, obtained from the above (rank-broken) pairwise estimates (see Lem. 15 in the proof of Thm. 12).

### G.3 Proofs for Thm. 12

**Theorem 12** (Regret bound of Sta′D++ (Alg. 4)). *Consider any $\delta > 0$, and suppose we set the parameters of Sta′D++ (Alg. 3) as $\eta = \frac{3}{2\kappa}\sqrt{2\log\frac{3qTK}{\delta}}$, and $t_0 = 2\left(\frac{C_1\sqrt{d} + C_2\sqrt{\log(2q/\delta)}}{\lambda_{\min}(B)}\right)^2 + \frac{4\Lambda}{\lambda_{\min}(B)}$, where $\Lambda$, $\kappa$, $B$ is as defined in Lem. 4. Then with probability at least $(1 - \delta)$, the $T$ round cumulative regret of Sta′D++ is at most $O\left(\frac{\sqrt{dT\log(T)}}{\kappa}\sqrt{\log\left(\frac{qTK}{\delta}\right)\log\left(\frac{Td}{\kappa}\log\frac{q}{\delta}\right)}\right)$.*

*Proof.* The main challenge in the regret analysis lies in justifying the right concentration rates of $\boldsymbol{\theta}_t^s$ computed from the MLE estimates of preferences in $\phi^s$. Towards this we use the result from [32]

---

**Algorithm 4 Sta′D++** (for general $q$-subsetwise preferences)

---

1: **input:** Learning rate $\eta > 0$, exploration length $t_0 > 0$
2: **init:** Select $t_0$ pairs $\{(\mathbf{x}_\tau, \mathbf{y}_\tau)\}_{\tau \in [t_0]}$, each drawn at random from $\mathcal{S}_\tau$, and observe the corresponding preference feedback $\{o_\tau\}_{\tau \in [t_0]}$
3: $S \leftarrow \lfloor \log T \rfloor, \phi^s \leftarrow \{(\mathbf{x}_\tau, \mathbf{y}_\tau, o_\tau)\}_{\tau \in [t_0]}, \ \forall s \in [\lfloor \log T \rfloor]$
4: Set $V_{t_0+1} := \sum_{\tau=1}^{t_0} (\mathbf{x}_\tau - \mathbf{y}_\tau)(\mathbf{x}_\tau - \mathbf{y}_\tau)^\top$
5: **while** $t \leq T$ **do**
6: $\quad s \leftarrow 1, \mathcal{G}^1 \leftarrow [K]$ $\hfill \triangleright$ Init stagewise pruning
7: $\quad$ **repeat**
8: $\quad\quad$ Compute the MLE estimate on $\phi^s$, i.e. solve for $\hat{\boldsymbol{\theta}}_t^s$ s.t.:
$$\sum_{(\mathbf{x}_\tau, \mathbf{y}_\tau, o_\tau) \in \phi^s} \left( o_\tau - \sigma\left((\mathbf{x}_\tau - \mathbf{y}_\tau)^\top \hat{\boldsymbol{\theta}}_t^s\right)\right)(\mathbf{x}_\tau - \mathbf{y}_\tau) = \mathbf{0}$$
9: $\quad\quad$ Set: $V_t^s = \sum_{(\mathbf{x}_\tau, \mathbf{y}_\tau) \in \phi^s}(\mathbf{x}_\tau - \mathbf{y}_\tau)(\mathbf{x}_\tau - \mathbf{y}_\tau)^\top$
10: $\quad\quad$ Compute: $g_t^s(i) = \hat{\boldsymbol{\theta}}_t^{s\top} \mathbf{x}_i^t, \forall i \in \mathcal{G}^s$, and $p_t^s(i,j) = \eta \|\mathbf{x}_i^t - \mathbf{x}_j^t\|_{V_t^{s-1}}, \forall i,j \in \mathcal{G}^s$
11: $\quad\quad$ **if** $p_t^s(i,j) \leq \frac{1}{\sqrt{T}}, \forall i,j \in \mathcal{G}^s$ **then**
12: $\quad\quad\quad$ $a_t \leftarrow \arg\max_{a \in \mathcal{G}^s} g_t^s(a)$. $\mathcal{X}_t \leftarrow a_t$
13: $\quad\quad\quad$ **for** $q' = 1, 2, \ldots (q-1)$ **do**
14: $\quad\quad\quad\quad$ $b_t \leftarrow \arg\max_{b \in \mathcal{G}^s \setminus \mathcal{X}_t} \left[ \min_{a \in \mathcal{X}_t} \left( g_t^s(b) + p_t^s(b,a)\right)\right]$ $\hfill \triangleright$ Max-Min subset selection
15: $\quad\quad\quad\quad$ $\mathcal{X}_t \leftarrow \mathcal{X}_t \cup \{b_t\}$
16: $\quad\quad\quad\quad$ **if** $\mathcal{G}^s \setminus \mathcal{X}_t = \emptyset$: Break the For Loop (Goto Line 18)
17: $\quad\quad\quad$ **end for**
18: $\quad\quad\quad$ Play $\mathcal{X}_t$. Receive $i_t \in \mathcal{X}_t$ $\hfill \triangleright$ Winner feedback of subset $\mathcal{X}_t$
19: $\quad\quad$ **else if** $p_t^s(i,j) \leq \frac{1}{2^s}, \forall i,j \in \mathcal{G}^s$ **then**
20: $\quad\quad\quad$ Find $\mathcal{B}_t^s := \{i \in \mathcal{G}^s \mid \exists j \in \mathcal{G}^s \text{ s.t. } g_t^s(i) + \frac{1}{2^s} < g_t^s(j)\}$
21: $\quad\quad\quad$ Update $\mathcal{G}^{s+1} \leftarrow \mathcal{G}^s \setminus \mathcal{B}^s, s \leftarrow s+1$
22: $\quad\quad$ **else**
23: $\quad\quad\quad$ Select a subset $\mathcal{X}_t \subseteq \mathcal{G}^s$ (up to size $q$) s.t. any pair $(a,b) \subseteq \mathcal{X}_t$ satisfies $p_t^s(a,b) > \frac{1}{2^s}$
24: $\quad\quad\quad$ Play $\mathcal{X}_t$. Receive $i_t \in \mathcal{X}_t$
25: $\quad\quad\quad$ Update $\phi^s \leftarrow \phi^s \cup \{(i_t, j, 1)\}_{j \in \mathcal{X}_t \setminus \{i_t\}}$ $\hfill \triangleright$ Converting $q$-subsetwise winner preferences to pairwise preferences: Rank-Breaking update
26: $\quad\quad$ **end if**
27: $\quad$ **until** a set $\mathcal{X}_t$ is found
28: $\quad$ Update: $t \leftarrow t+1$
29: **end while**

---

which shows that for Plackett-Luce model, the rank broken pairwise estimates are unbiased for true pairwise preferences (see Lem. 1, [32]).

Now recall that our subsetwise preference feedback model also corresponds to a Plackett-Luce model with utility parameters of item $\mathbf{x} \in \mathcal{D}$ being $e^{\mathbf{x}^\top \boldsymbol{\theta}^*}$ (as discussed in Sec. 2.1). Hence above result applies to our setup which implies for any triplet $(\mathbf{x}_\tau, \mathbf{y}_\tau, o_\tau)$ in $\phi^s$, we have $o_\tau \sim \text{Ber}\left(\sigma((\mathbf{x}_\tau - \mathbf{y}_\tau)^\top \boldsymbol{\theta}^*)\right)$. Consequently, we can hence again apply the same *finite-samples-asymptotic-normality for MLE estimates of GLM models* (Thm. 1,[26]) to derive a similar *sharper concentration of pairwise scores* as used in Lem. 4 for the original Stagewise-Adaptive-Duel algorithm (Alg. 3) for pairwise preferences. Precisely, in this case we can derive the following concentration bound:

**Lemma 15.** *Consider any $\delta > 0$, and suppose we set the parameters of Sta′D++ (Alg. 3) as* $\eta = \frac{3}{2\kappa}\sqrt{2\log\left(\frac{3qTK}{\delta}\right)}$, *where* $\kappa := \inf_{\|x-y\| \leq 2, \|\boldsymbol{\theta}^* - \hat{\boldsymbol{\theta}}\| \leq 1} \left[\sigma'((\mathbf{x} - \mathbf{y})^\top \hat{\boldsymbol{\theta}})\right]$, *and* $t_0 = 2\left(\frac{C_1\sqrt{d} + C_2\sqrt{\log(2q/\delta)}}{\lambda_{\min}(B)}\right)^2 + \frac{4\Lambda}{\lambda_{\min}(B)}$, *where* $\Lambda = \frac{8}{\kappa^4}\left(d^2 + \log(3q/\delta)\right)$ *and* $B = \mathbf{E}_{\mathbf{x},\mathbf{y} \overset{iid}{\sim} \mathcal{P}_{\mathcal{D}}}[(\mathbf{x} - \mathbf{y})(\mathbf{x} - \mathbf{y})^\top]$ *(for some universal problem independent constants $C_1, C_2 > 0$). Then with probability*

*at least* $(1 - \delta)$, *for all stages* $s \in \lceil \log T \rceil$ *at all rounds* $t > t_0$ *and for all index pairs* $i, j \in \mathcal{G}^s$ *of round* $t$: $|(\mathbf{x}_i^t - \mathbf{x}_j^t)^\top (\boldsymbol{\theta}^* - \boldsymbol{\theta}_t^s) \leq p_t^s(i,j)|$.

As shown in the proof of Thm. 5, this concentration guarantee is really the key result used towards proving the regret bound of Alg. 3. It is easy to check that, given above concentration, the other two supporting lemmas (Lem. 6 and Lem. 7) can easily be shown to hold good in this case as well due to the similar 'pairwise-preference' based arm-selection strategy as that of Alg. 3 (with some additional care needed to incorporate the *max-min subset selection* strategy used for this case, same as the technique suggested in [33]). Moreover Lem. 14 simply follows for this algorithm as well since we maintain independent pairwise samples in $\phi^s$ across different stages $s \in \left[ \lfloor \log T \rfloor \right]$ (same as Alg. 3). The result of Thm. 12 can now be derived by combining the above results, similar to argument shown in the proof of Thm. 5. $\qquad \square$