# OpenReview forum: "Optimal Algorithms for Stochastic Contextual Preference Bandits "
_NeurIPS.cc/2021/Conference — NeurIPS 2021 Poster_

### Official Review · Reviewer_TXKE · 2021-07-16

**Rating:** 5
**Confidence:** 3

**Summary:**

This paper studies the problem of preference bandits in the contextual setting, where the online learner plays $q$ arms and observes a noisy winner of the subset. The goal is to identify the best arm given a context. Two special algorithms for the $q = 2$ case are presented, along with the regret upper bound and a matching lower bound. The results are then extended to a general $q$. Corresponding experiments are presented to verify the theoretical results.

**Ethical Concerns:**

N.A.

**Limitations And Societal Impact:**

See comments on limitations above. There is no immediate societal impact.

**Main Review:**

This paper presents two novel algorithms for the stochastic contextual preference bandits for the special case $q = 2$, and extends the algorithm and corresponding regret upper and lower bound to more general $q$.

Main comments on the clarity, quality, and significance:

1) The motivation does not match with the problem: The introduction motivates the problem by an application in recommender systems, while the proposed algorithm needs to play repeated arms in one step (as pointed out in lines 102-104). The motivation is therefore slightly confusing as the same product or advertisement typically will not be shown repeatedly in one step. The linear MNL model (e.g., by Oh and Iyengar, 2019) seems to be a better fit for the motivation example. It is slightly hard to appreciate the significance without proper motivation.

2) Comments on the lower bound (Theorem 10): The reduction from linear-contextual bandits to the studied framework is neat. It would be better to clarify the relation of regret on $\mathcal{I}^{clb}$ and $\mathcal{I}^{cdb}$ in Lemma 9. The current statement of "algorithm runs on a problem instance" is slightly vague and confusing (and the "rums" seems to be a typo...)

3) The results on the general $g \ge 2$ case: While the most interesting results of this paper seem to be the algorithm/analysis for the general setting (i.e., $g \ge 2$), most related discussions are deferred to appendix. Given that the special case of $g=2$ (discussed from page 3 to page 7) is covered by the general case, please consider compressing the special case of $g=2$.

3) The experiment results: In Figure 4, it is hard to see the down-going trend of performance when increasing the dimensionality (espeically given the error bar is so small). Maybe a larger range of $d$ will show a clear pattern. It is also hard to go through the experiment results with the socre functions $h(d, K)$, $e(d, K)$ and $m(d,K)$ are all in appendix. Please consider bringing them back.

4) Minor comments on writing: The writing can be largely improved. Besides the comments above -

a) Line 182, Lemma 1, there is a ")" missing after $y_2$.

b) Line 283, "attend" -> "attain".

c) Line 360, Remark 2: "algorithms still outperforms" -> "algorithms still outperform others"

d) Line 344, (q) should be boldfaced.

e) Figure 2 caption is covered by Figure 4.

f) Figure 3 caption: "Set size (K)" -> "Context-size (K)"?

**Time Spent Reviewing:**

5

---

> ### Author Response · Authors · 2021-08-09
> **Response to Reviewer TXKE**
>
> Thanks for your careful reviews and insightful comments.
>
> Re. Motivation: Our problem motivation is borrowed from regret minimization in preference bandits literature (e.g. [3],[33],[44],[47], Sui et al'18, Bengs et al'21 etc.) -- Like the well-studied regret objective in dueling bandits as used in the references above, we model applications where relative feedback is easier/more natural to obtain compared to the absolute reward. An example motivating regret with preferences is: Consider the problem of recommending items from a catalog to users on a shopping website. Each time, the context is determined by the visiting user’s features together with all items’ features. When a pair of items is presented, the user clicks on one or the other according to a relative preference model where only the items presented matter---this plausibly models comparative cognitive choices being made by humans. Suppose there is a separate utility for the clicked-upon item (like ad revenue) that is accumulated in parallel, but with the learner only observing the (coarse) preferential decision. The aim is to converge to determine the overall best item in the catalog, optimizing the net (financial) reward earned by the system.
>
> Re. elaborating Lem 9: We would make the description clearer and also include the statement “2R^{cdb}_T = R_^{clb}_T” in the lemma statement. To clarify it in words: If the clb-algorithm A^{clb} is receiving feedback from a K-armed (stochastic) contextual linear bandit instance parameterized by the unknown d-dimensional vector \theta^* for 2T rounds, then, in turn, the underlying dueling bandit algorithm A^{cdb} receives feedback from a K-armed contextual linear dueling instance parameterized by \theta* for T rounds (since two successive rounds of CLB comprises one round of CDB by the reduction step), and the regret incurred by A^{cdb} is exactly half of the regret incurred by A^{clb}. [Thanks for catching the typo: "rums"]
>
> Re. discussion on the general setup (q>=2): Thanks for the suggestions, we thought of elaborating the special (q=2) case first to explain our main algorithm ideas for learning from preference feedback in the stochastic contextual setting and later generalized it to any general q-subsetwise feedback setup to clearly compare the regret tradeoff with subsetsize-q. But we agree that Sec 5 requires more discussion space, will elaborate that in the update (reducing the space for Sec 3 by removing some of the lemmas and description text).
>
> Re. plotting the results better: Our algorithms consistently perform better than the baselines even for higher ranges of d, if advised, will be happy to include these results (higher d) in the update.
>
> Thanks a lot for pointing the typos, will correct all of them (and proofread thoroughly) in the updated version.
>
> We request the reviewer to kindly reconsider the final score based on the responses above.

---

> > ### Author Response · Authors · 2021-09-11
> > **Post rebuttal**
> >
> > Dear Reviewer TXKE,
> >
> > We hope that our responses have adequately addressed your concerns. We would greatly appreciate it if you please reconsider the scores based on that. Of course, we would be happy to clarify if there is any other concern.
> >
> > Thanks
> > Authors

---

### Official Review · Reviewer_tuGP · 2021-07-18

**Rating:** 6
**Confidence:** 4

**Summary:**

The authors study the contextual bandit problem where the feedback is received in the form of preferences. In this setting there are K arms and at each time step k random context vectors x^t_1, .. x^t_K are drawn i.i.d. From some distribution. The algorithm can output a subset of arms of size q and gets to see a noisy version of which of the q arms has the highest reward. This noisy distribution is assumed to be the Plackett-Luce model (a popular ranking model) with parameter theta*. The goal is to compete with the regret of the best arm (assuming expected reward of arm i is theta* \cdot x^t_i). The reward incurred by the algorithm is assumed to be the average reward of the subset played at time t.

**Ethical Concerns:**

None.

**Limitations And Societal Impact:**

None.

**Main Review:**

The authors propose a simpler algorithm that achieves O(d sqrt{T}) regret and a more sophisticated stagewise algorithm that achieves the optimal O(sqrt{dT}) regret. The authors also show a matching sqrt{d T} regret for the case of general q, thereby showing that the stagewise algorithm is near optimal.

The problem is well motivated and the authors provide optimal regret guarantees for the proposed formulation. Although the techniques are inspired by existing work on contextual bandits, non-trivial ideas are needed to extend them to the preference setting. It is not clear why and how the algorithm proposed in section 5.2 is adding anything to the results given that the stage-wise algorithm already is optimal. Furthermore, it is not clear if the lower bounds proved in the paper indeed apply to the contexts being generated i.i.d. from some distribution.

**Time Spent Reviewing:**

1.5

---

> ### Author Response · Authors · 2021-08-09
> **Response to Reviewer tuGP**
>
> Thanks for taking time reviewing our paper.
>
> Re. usefulness of the algorithm in Sec. 5.2: It’s correct that our Alg. 2 (Sta’D) gives the optimal bound when q=2, but, as motivated in Line 285-195, Sta’D++ works produces the optimal regret guarantee (matching the regret lower bound of Thm11) even when the learner is restricted to play fixed subsets of size q>2 (and not any subsets of sizes 1,2,...,q), which Sta’D is designed to work with. Moreover, in future, we hope to analyze a more optimal gap-dependent regret-bound for both and upper and lower bounds (as opposed to worst-case gap-independent bound) which will bring out a clearer dependency on regret-vs-q for any general fixed (q>=2).
>
> Re. if lower bound applies to iid contexts: Yes, it does. Please note, as discussed in Sec 4, Lem 8 shows that any stochastic linear bandit problem can be reduced to our contextual dueling problem with regret equivalence (up to constant factors of 2). To clarify it in words: If the clb-algorithm A^{clb} is receiving feedback from a K-armed (stochastic) contextual linear bandit instance parameterized by the unknown d-dimensional vector \theta^* for 2T rounds, then, in turn, the underlying dueling bandit algorithm A^{cdb} receives feedback from a K-armed contextual linear dueling instance parameterized by \theta* for T rounds (since two successive rounds of CLB comprises one round of CDB by the reduction step), and the regret incurred by A^{cdb} is exactly half of the regret incurred by A^{clb} (i.e. 2R^{cdb}_T = R_^{clb}_T). And hence regret lower bound for any stochastic linear bandits problem with iid contexts in turn implies a lower bound for our problem setup (with iid contexts), e.g. see Lem7 of [13] (as referred in Line 268) which gives the lower bound for stochastic linear bandits with iid contexts, precisely Omega(\sqrt{dT}).

---

### Official Review · Reviewer_K3kQ · 2021-07-21

**Rating:** 7
**Confidence:** 3

**Summary:**

This paper studies the following learning problem. A learner wishes to learn some fixed preference vector theta in R^d. Every round (for T rounds), a set of K d-dimensional actions S_t are sampled. The learner must choose a subset (possibly of bounded size) X of these actions. The learner then learns a noisy signal of which of the actions in the subset they chose is optimal, and receives as reward the average reward of the actions in X (where the reward of action x is simply <x, theta>).

The paper first studies the case where the learner is only allowed to select subsets of size 2 (the “dueling bandits” setting). The authors give two algorithms for this problem. The first algorithm they give guarantees regret ~O(d*sqrt(T)). This algorithm works by choosing the “maximum-informative” pair of actions each round (i.e. the pair of actions for which the estimator of which action will win has highest variance). They then give a second algorithm which reduces the dependence on d to sqrt(d) at the cost of an additional sqrt(log K) term (for a total runtime of O(sqrt(d log T log K)). At a high level, this algorithm is similar to the previous algorithm but operates in a sequence of ~log T stages, eliminating some arms each round. Finally, the authors demonstrate an Omega(sqrt(d T)) lower bound via a reduction from linear-contextual bandits.

The authors show how to extend these results to the case where the learner may choose a subset of more than 2 actions. Interestingly, this does not seem to significantly help; even with this ability, there is still an Omega(sqrt(dT)) lower bound. Moreover, even if the learner is only allowed to choose subsets of size q or greater, it is still possible to modify the algorithms above to get ~O(sqrt(dT)) regret (the main idea being that it is still possible to solicit comparisons between two items even if you are forced to use larger action sets).

Finally, the authors evaluate their algorithms empirically on some synthetic data sets, showing that they perform better than existing dueling algorithms for this setting.

**Limitations And Societal Impact:**

No concerns here.

**Main Review:**


I enjoyed reading this paper -- I think the setting the authors study is quite natural, and their results are fairly comprehensive (e.g. almost matching upper and lower bounds). Technically, while none of the techniques are extemely novel, the analysis is certainly non-trivial. The paper was well-written and easy to read.

One question I had after reading this paper: do any of the algorithms extend to the setting where the set of K active arms is chosen by an adversary every round instead of being sampled stochastically? My guess is that it messes with the concentration bounds in the analysis (e.g. Lemma 4), but I haven’t thought about it much. It seems like a natural setting to explore.


**Time Spent Reviewing:**

3

---

> ### Author Response · Authors · 2021-08-09
> **Response to Reviewer K3kQ**
>
> Many thanks for your careful reviews and appreciating our work.
>
> Re. “adversarially chosen arms”: Great question. Indeed (as the title suggests) our methods are only applicable to stochastic contexts as the algorithmic ideas are based on estimating the rewards (i.e. linear-score estimates) with high probability (Lem 2, 4 etc.) which is not possible for adversarial contexts. To the best of our knowledge, there are no efficient+optimal regret algorithms for “adversarial subsetwise-preference bandits” which requires the incorporation of adversarial methods like EXP4 (see Ref. [15]). Exploring this certainly is an active line of research and very much lies in the direction of our future interests.

---

### Decision · Program_Chairs · 2021-09-27

**Decision:**

Accept (Poster)

**Comment:**

This paper studies the problem of k-armed contextual preference bandits and gives optimal efficient algorithms matching lower bound regret guarantees.  Empirical studies are also included.  The results appear interesting and novel.  Despite some concerns about the motivation, the paper makes a sufficient contribution overall to warrant acceptance.